# Establishment of a Cell Culture Model of Persistent Flaviviral Infection: Usutu Virus Shows Sustained Replication during Passages and Resistance to Extinction by Antiviral Nucleosides

**DOI:** 10.3390/v11060560

**Published:** 2019-06-17

**Authors:** Raquel Navarro Sempere, Armando Arias

**Affiliations:** 1Life Science & Bioengineering Building, Technical University of Denmark, 2800 Kongens Lyngby, Denmark; rnsempere@abiopep.com; 2Abiopep Sociedad Limitada, Parque Científico de Murcia, 30100 Murcia, Spain

**Keywords:** chronic viral infection, emerging arboviruses, defective viral genomes, antiviral therapies, lethal mutagenesis

## Abstract

Chronic viral disease constitutes a major global health problem, with several hundred million people affected and an associated elevated number of deaths. An increasing number of disorders caused by human flaviviruses are related to their capacity to establish a persistent infection. Here we show that Usutu virus (USUV), an emerging zoonotic flavivirus linked to sporadic neurologic disease in humans, can establish a persistent infection in cell culture. Two independent lineages of Vero cells surviving USUV lytic infection were cultured over 82 days (41 cell transfers) without any apparent cytopathology crisis associated. We found elevated titers in the supernatant of these cells, with modest fluctuations during passages but no overall tendency towards increased or decreased infectivity. In addition to full-length genomes, viral RNA isolated from these cells at passage 40 revealed the presence of defective genomes, containing different deletions at the 5’ end. These truncated transcripts were all predicted to encode shorter polyprotein products lacking membrane and envelope structural proteins, and most of non-structural protein 1. Treatment with different broad-range antiviral nucleosides revealed that USUV is sensitive to these compounds in the context of a persistent infection, in agreement with previous observations during lytic infections. The exposure of infected cells to prolonged treatment (10 days) with favipiravir and/or ribavirin resulted in the complete clearance of infectivity in the cellular supernatants (decrease of ~5 log_10_ in virus titers and RNA levels), although modest changes in intracellular viral RNA levels were recorded (<2 log_10_ decrease). Drug withdrawal after treatment day 10 resulted in a relapse in virus titers. These results encourage the use of persistently-infected cultures as a surrogate system in the identification of improved antivirals against flaviviral chronic disease.

## 1. Introduction

Chronic viral infections constitute a major challenge to global public health, with several hundred million people affected and significant associated fatalities [1,2,3,4,5,6]. Viral persistence in the host generally leads to different pathogenic outcomes including the exacerbation of medical conditions [7]. Apart from their direct relationship with disease, viral persistence in animal reservoirs has also been linked to the re-emergence of pathogenic viruses [7]. In addition to major viral agents causing chronic infections, such as hepatitis C virus (HCV) and human immunodeficiency virus [3,8], there is an increasing incidence of chronic disease caused by arboviruses [6,9,10,11]. In the flaviviruses, the ability to persist seems to be limited to neurotropic strains. However, in addition to neurological infection, it has been documented that flavivirus persistence also occur in different cells and tissues outside the nervous system [6]. In particular, West Nile virus (WNV) has been isolated from the urine of patients with a history of infection and suffering from chronic renal disorders [12,13]. Further evidence supporting an association between chronic disease and persistent kidney infection has been obtained with animal models for WNV and other flaviviruses [12,14,15,16,17]. Besides severe neurological disorders linked to acute infection (e.g., Guillain–Barré in adults and congenital brain defects in neonates), Zika virus (ZIKV) establishes persistence in different cells and tissues, leading to an array of other medical conditions [9,18,19]. Persistent ZIKV in both male and female reproductive tracts is connected to sexual transmission of infection [18,20], and also to mother-to-fetus transmission from vaginal infected tissue [18,21]. Tick-borne encephalitis virus (TBEV) and Japanese encephalitis virus (JEV) also cause chronic disease associated with persistence. Progressive chronic TBEV was documented in a patient who died ten years after infection [22], while JEV was detected in the cerebrospinal fluid of infected patients for several weeks to >100 days after the onset of symptoms [23]. The ability of flaviviruses to persist is not limited to their vertebrate hosts as they can also infect their vectors for long periods. WNV can be detected in different mosquito tissues for at least 4 weeks after infection [24,25] while TBEV replicates in ticks for periods exceeding 100 days [26]. The life of ticks can be several years which could potentially help to propagate persistent flaviviruses in their vertebrate hosts multiple times during a lifetime.

While most flaviviruses generally trigger apoptotic cell death in vitro, many of them can establish a persistent infection in different cultured cells. Specifically, TBEV can persist in the human embryonic kidney (HEK) 293T cell line for at least 35 weeks without any cytopathic effect crisis observed [27,28]. Viral RNA levels remained constant during the whole period although virus titers decayed after four weeks. This drop in infectivity was linked to the emergence of defective interfering (DI) particles, although the establishment of persistence could not be directly related to them. TBEV defective genomes presented a deletion partially affecting envelope (Env) and non-structural protein 1 (NS1) genes [27]. Transcriptome analysis of these persistently-infected cells revealed increased expression of survival genes and downregulation of apoptosis-associated factors [28]. Cell models of flaviviral persistence have also been established in insect cells [29,30]. TBEV persistence in different tick cell lines led to similar evolutionary trends as those observed in HEK cells, with virus titers detected for >100 days, a tendency to lower plaque sizes, and different adaptive mutations in the Env gene, although no DI particles were found [26].

Cell culture lineages persistently infected with viruses represent tantalizing models in the study of chronic infection in vitro and the characterization of antiviral drugs [31,32]. Most culture models for flaviviruses typically involve a lytic infection, leading to the death of the cell monolayer. However, these systems may be less adequate for examining the efficacy of inhibitors that need to be administered during a prolonged period. For HCV, a virus genetically related to arthropod-transmitted flaviviruses (both belong to the *Flaviviridae* family), there have been developed cell culture systems of persistent infection which bear some degree of resemblance to chronic infection in patients [32,33,34]. Tissue culture models for persistent HCV have replicated the efficacy of multidrug therapies currently used in the clinical treatment of infection, including sofosbuvir [32]. These precedents encourage the establishment of cell culture methods for different viruses, such as the flaviviruses, as possible surrogate models to examine persistence and to identify drugs with improved efficacy.

In this study, we have investigated the capacity of Usutu virus (USUV) to establish persistence in African green monkey kidney epithelial cells (Vero cells). These cells are permissive to most human flaviviruses, and are generally employed in their propagation and biological characterization. USUV is an emerging threat, rapidly spreading in different wild and captive birds in Europe, and causing large mortality in some species, e.g. blackbirds [35,36]. Although USUV infection in humans is typically asymptomatic, there has been reported an increasing number of neurological cases resembling WNV infection. The incidence of USUV infection in humans seems to be on the rise, with a record number of infected people in Austria during 2018 [37,38]. There is evidence suggesting that increased incidence of disease in humans is geographically connected to outbreaks in birds [39,40,41,42,43]. Besides, human cases of USUV disease might have been historically misdiagnosed as WNV, underlining the relevance of this pathogen as an emerging threat to global health, and the need for new tools for its study [39,40,44]. Owing to its close relationship to WNV, which is known to cause chronic infection in humans, it is conceivable that USUV can also establish persistence in its hosts.

Here, we have isolated two independent lines of Vero cells that became persistently infected after surviving an apparently lytic infection with USUV. The advantage of including two cellular lineages is that those biological features reproduced in both lines may be more relevant to understanding viral persistence. We showed that persistently-infected cells were successfully maintained for 82 days (41 passages), with viral infectivity positively detected in all the cellular supernatants. The genetic analysis of viral RNA extracted from the cells after 40 passages revealed the presence of both full-length and defective viral genomes in each cell line. Different in-frame deletions (>2 Kb) were identified, all of them located at the 5’ end of the viral RNA. These truncated genomes are predicted to encode shorter viral polyproteins lacking partially or entirely several structural proteins (membrane (M), its precursor (PrM), and envelope (Env) proteins) and non-structural protein 1 (NS1).

To calibrate the efficacy of this cell culture model in the study of antiviral compounds, we have used three broad-range nucleoside analogues that we previously examined during a lytic infection [45]. In our earlier work, we demonstrated that ribavirin (RBV), favipiravir (FAV), and 5-fluorouracil (FU) elicit a strong antiviral activity associated with viral mutagenesis [45]. Under certain experimental conditions, we observed complete virus extinction after five consecutive lytic passages in the presence of these drugs [45]. Additional in vivo evidence of the antiviral efficacy of FAV against USUV has been obtained by Segura Guerrero and colleagues using a mouse model of lethal infection [46]. In this study, we found that these compounds also exhibit antiviral activity against persistent USUV. Prolonged exposure to RBV, FAV, or a combination of both drugs (F + R) can lead to the complete extinction of infectivity and viral RNA in the cellular supernatants. However, significant amounts of intracellular viral RNA were yet detected in these cells, suggesting that USUV escapes extinction by mutagenesis in a persistently-infected cell context. We discuss the possible value of persistently-infected cell models for the identification of new antiviral compounds against the flaviviruses. We posit that these methods may permit the characterization of antiviral activity during extended treatment periods, and thus be instrumental to the identification of improved drugs against chronic infection.

## 2. Materials and Methods

### 2.1. Cells, Virus, and Establishment of Persistently Infected Cultures

To this study we used an USUV strain isolated from infected birds in Austria (2001) and provided by Giovanni Savini from Istituto G. Caporale, Italy [45,47]. The viral stock used for this study was obtained after seven USUV passages in cell culture. For propagation and titration of viral samples we employed Vero cells as previously described [45]. Cells were grown in media containing 5% (v/v) fetal bovine serum (FBS, Sigma, St. Louis, MO, USA), 100 units/mL penicillin–streptomycin (ThermoFisher, Waltham, MA, USA) and 1 mM Hepes in high glucose DMEM (ThermoFisher), and maintained at 37 °C with 5% CO_2_.

For the establishment of persistently-infected cultures we seeded 1 × 10^6^ Vero cells in 35 mm culture dishes, and incubated them overnight at 37 °C in the presence of 5% CO_2_. We then removed the cellular supernatant and inoculated USUV at a multiplicity of infection (MOI) of 0.01. Extensive cytopathic effect was observed at 48 h post-infection although we found that ~20% of the cells remained attached to the plate even at later post-infection times (Figure 1). At 48 (line V) or 96 h post-inoculation (line S), unattached dead cells were removed by washing the plate three times (using fresh media or PBS). Those cells surviving the infection were supplemented with fresh media and, after reaching confluence, trypsinized and seeded in a new flask. The established cell lines were then passaged every 48 h by transferring 3–4 × 10^6^ cells to a new 75 cm^2^ flask in each passage.

### 2.2. Virus Titration

To analyze virus titers, cellular supernatants were collected from persistently-infected cells (at 48 h post-seeding) before each cell passage. The presence of infectious virus in these supernatants was determined by 50% tissue culture infectious dose (TCID_50_) assays as previously described [45]. Briefly, to each well of a 96-well plate 1 × 10^4^ Vero cells in 60 µL of media containing 5% FBS was added. On the following day, 100 µL of 10-fold serial dilutions of each viral sample, in media containing 1% FBS, was applied to each well, reaching a final volume of 160 µL. The virus titers were determined by scoring the number of infected wells showing apparent cytopathic effect at day 5 post-infection, and using the Reed and Muench method [48,49].

### 2.3. Treatment with Antiviral Compounds

To determine the 50% inhibitory concentration (IC_50_) for each drug treatment, we seeded 2 × 10^4^ cells per well in a 96-well plate. On the following day, cell supernatants were removed and 100 µL of 1% FBS complete media containing 5-fluorouracil (2,4-dihydroxy-5-fluoropyrimidine, Sigma-Aldrich), ribavirin (1-(β-d-Ribofuranosyl)-1H-1,2,4-triazole-3-carboxamide, Sigma-Aldrich), favipiravir (6-fluoro-3-hydroxy-2-pyrazinecarboxamide, Atomax) or a combination of ribavirin and favipiravir at concentrations ranging from 25 to 2000 µM was added. Cells were incubated at 37 °C and in the presence of 5% CO_2_, and supernatants containing virus were collected at different time points for subsequent analyses.

Cell toxicity assays were performed on 96-well plates using the same drugs and concentrations abovementioned for IC_50_ assays. Toxicity was recorded using the CellTiter–Blue Cell viability assay (Promega, Madison, WI, USA), which accounts for living active cells, following the indications provided by the manufacturer.

To examine the effect of antiviral nucleosides upon persistent USUV during prolonged treatment, we seeded 1 × 10^5^ cells per well in a 24-well plate. On the following day, cell culture supernatants were removed and 1 ml of 1% FBS complete media containing 2000 µM ribavirin, favipiravir, or a combination of ribavirin and favipiravir were added. Cells were incubated at 37 °C and in the presence of 5% CO_2_.

### 2.4. Viral RNA Extraction, RT-PCR Amplification, and Sequence Analysis

Viral RNA was extracted from either 100 µL of cell supernatants or directly from whole cell monolayers grown in 24-well plates, using GeneJET RNA Purification Kit (ThermoFisher). To amplify the USUV viral RNA molecules, we followed a two-step RT-PCR amplification procedure. Briefly, 4 µL of purified RNA extracted from whole cell monolayers were reverse-transcribed in a final volume of 20 µL using SuperScriptIII (Roche, Basel, Switzerland), as indicated by the manufacturer. Three microliters of cDNA were then PCR amplified using AccuPrime Taq DNA Polymerase High Fidelity as indicated by the provider (Thermofisher). To generate overlapping amplicons covering the entire USUV genome, we used primers spanning genomic positions: 1 to 55 (sense) and 7950 to 7929; 4218 to 4189; 3359 to 3335 or 2673 to 2653 (antisense); 818 to 842 (sense) and 3359 to 3335 (antisense); 2976 to 3009 (sense) and 7950 to 7929 (antisense); and 6841 to 6864 (sense) and 11066 to 11016 (antisense). PCR products were directly extracted with a QIAquick PCR Purification Kit (Qiagen, Hilden, Germany) or individual PCR bands on an agarose gel were purified using the PureLink Quick Gel Extraction Kit (Invitrogen, Carlsbad, CA, USA). The resulting PCR products were Sanger sequenced (LGC Genomics, Berlin, Germany). Full genome sequences of our stock virus (passage 0) or USUV adapted to persistence (S and V cells at passage 40) were compared to the reference strain (Genbank ID: AY453411) using the CLC Main Workbench program [47].

### 2.5. Quantitative PCR Analysis of Virus Populations

To detect total viral RNA in the sample (including both standard and defective genomes), we used a protocol previously described [45,50]. Briefly, in this assay we used sense and antisense primers, and a FAM–TAMRA probe targeting the NS5 gene (position 9297 to 9318). For the amplification of the RNA sample we used TaqMan Fast Virus 1-Step Master Mix (ThermoFisher) and one-step RT-PCR amplification conditions, with a reverse-transcription step (30 min at 48 °C) followed by 1 min incubation at 95 °C and 40 amplification cycles of 15 s at 95 °C and 1 min at 60 °C. For the specific quantification of full-length genomes (excluding defective genomes lacking Env) we used primers targeting the Env protein-coding gene spanning residues 1462 to 1486 (sense) and 1559 to 1540 (antisense). For this method we used One-Step TB Green PrimeScript RT-PCR Kit II (Takara, Tokyo, Japan) and one-step RT-PCR amplification conditions, with a reverse-transcription step (5 min at 42 °C) followed by 2 min incubation at 95 °C and 40 amplification cycles of 15 s at 95 °C and 30 s at 60 °C.

To obtain a standard curve of known amounts of viral genomes, samples were prepared with USUV RNA extracted from lytically-infected cells. To quantify the number of viral RNA molecules present in the extract we used known amounts of USUV cDNA (plasmid containing USUV genomic positions 8965 to 10,398) and the FAM–TAMRA method abovementioned. Once viral genome concentration was determined, the reference USUV RNA used to prepare the standard curve was distributed in aliquots of 5–10 µL in low-biding tubes and stored at −80 °C. For each qPCR experiment we used fresh aliquots (only frozen–thawed once) and discarded them after the experiment. The standard curve samples were prepared on the same day as the qPCR assay by diluting our reference USUV RNA to a final concentration of 10^9^ viral RNA molecules/µL in 100 ng/µL yeast RNA. This sample was serially diluted 1:10 in yeast RNA to generate aliquots containing 10^8^–10^0^ molecules/µL. Two independent sets of standards were prepared for each qPCR plate.

### 2.6. Statistical Analysis

Statistical significance was determined using GraphPad Prism 8 (LaJolla, CA, USA) as specified in each corresponding figure legend.

## 3. Results

### 3.1. Establishment of Cell Lines Persistently Infected with USUV

To establish persistently-infected cultures we inoculated USUV onto two monolayers of Vero cells that are subsequently referred as V and S lines (Figure 1). At 48 h post-infection extensive cytopathic effect was observed, with an estimated ~80% of cells detaching from the plate (Figure 1B). To remove detached dead cells, we extensively washed the plates with fresh media at 48 h (lineage V) or 96 h (lineage S). Cell monolayers surviving the infection were cultured normally and, after reaching confluence, the monolayer was trypsinized and transferred to a new flask using standard procedures for the passage of non-infected cells. V and S cells exhibited normal growth, modestly slower than Vero, leading to confluent monolayers of similar appearance (Appendix A). V and S cell lines were maintained for 82 days (41 transfers) by serially passaging them every second day (Figure 1). No apparent lytic crises were observed during the passages.

To investigate whether the surviving cells had become infected, we analyzed the presence of infectious virus (TCID_50_ assay) and viral RNA (qPCR) in the cellular supernatants. Both cell lines showed elevated titers along the whole passage history, suggesting that infectious USUV is continuously shed by these cells to the extracellular environments. Notably, cellular supernatants collected from persistently-infected cells (which showed no cytopathology) caused normal cytopathic effects during infection on regular Vero cell cultures, permitting virus titration as normal. This observation may be indicative of V and S cells becoming resistant to lytic infection. Viral titers fluctuated between 10^5^ and 10^8^ TCID_50_/mL although no overall tendency to higher or lower virus yields were observed during the passages (*m* is 0.01 and 0.00 for cell lines V and S, respectively). USUV specific infectivity, which measures the ratio between infectious virus particles and viral genomes, was decreasing along passages (Figure 1E,F), with larger viral RNA levels detected in the supernatant of later samples (Figure 1D). This might be indicative of USUV infectivity losses associated with long-term persistence in cell culture, in agreement with previous observations for other flaviviruses [27].

To characterize the genetic alterations associated with long-term persistence, we sequenced the entire USUV genome isolated from persistently-infected cells after 80 days (passage 40). Different PCR amplicons covering the USUV 5’-end genome revealed the presence of large deletions in both V and S cells (Figure 2). In V cells, we identified three bands of different sizes in amplifications of the 5’ end region, one corresponding to the expected length and two other bands of lower molecular weight. Further sequence analysis confirmed the presence of two in-frame deletions in these amplicons of 2.5 and 2.1 Kb (Δ585–3053, Δ872–2971). In S cells, we detected a single dominant band ~2.5 Kb shorter than the expected size in each of several PCR amplifications of 5’ USUV RNA (Figure 2B, see amplicons I, III, and IV). Sequence analysis of these samples revealed the presence of two major in-frame deletions of similar size, spanning residues 584 to 3184 (2.6 Kb) and 680 to 3121 (2.4 Kb), which may account for this apparent single band. Standard genomes were detected in S cells only when primers annealing to the deleted region were used (Figure 2B, PCR amplicons V and VI), suggesting that the absence of full-length genome products in PCRs I, III, and IV may be due to a preferential amplification of truncated variants. All these deletions were predicted to render in-frame truncations of the polyprotein lacking the Env protein and most of NS1 (Figure 2). In three of these four truncated genomes the complete loss of structural M protein was also predicted.

### 3.2. Quantification of Truncated and Standard Viral Genomes in Persistently-Infected Cells

To quantify the relative amount of full-length genomic RNA to total viral RNA (including both standard and defective genomes) we used two different qPCR quantification methods specifically targeting two genomic regions: Env and NS5 (Figure 3). The qPCR approach for the detection of the Env gene (absent in defective viral RNA molecules) is predicted to identify standard genomes only. As anticipated, we found lower amounts of viral RNA in V and S cells when using qPCR methods for Env than for NS5. No significant differences were obtained for viral RNA extracted from cells infected with wild type USUV (Figure 3B,C), in agreement with the absence of defective genomes in this sample (Figure 2B,C, sample p1). From this analysis we estimate that 71% of viral genomes in S p40 cells are truncated (29% of standard genomes) while in V p40 cells 46% of genomes are defective (54% of standard genomes). This is in agreement with better PCR detection of full-length genomes in samples isolated from V cells than in S cells (Figure 2B,C, amplicons I, III, IV).

The quantitative analysis of viral RNA in cellular supernatants also suggests the release of particles containing truncated genomes to the extracellular environment, as lower proportions of standard USUV genomes are detected in S and V cells than in Vero cells infected with wild type USUV (Figure 3D,E, Appendix A). The analysis of extracellular samples from different passages also supports the presence of defective genomes starting from an early passage in both V and S lines (Figure 3F, Appendix A).

### 3.3. Non-Synonymous Mutations are Accumulated in USUV Proteins NS4A and NS4B

Full-length genome sequencing of viral RNA isolated at passage 40 revealed the presence of four repeated nucleotide substitutions in both V and S cell lineages. These changes included two silent mutations in the Env and M coding regions, and two non-synonymous mutations in NS4A and NS4B-coding regions, leading to amino acid replacements E45N and T92M, respectively (Table 1). Another amino acid replacement in NS4A (F24L) was also found in both lines, although it was the result of different nucleotide changes (Table 1). In addition to these three amino acid replacements repeatedly found in both viruses, other changes were identified in these two proteins (E22D in NS4A; L89R, Y127H and M172R in NS4B), supporting an adaptive role to persistence in cell culture.

### 3.4. Broad-Range Nucleoside Drugs Strongly Inhibit Persistent USUV

To characterize this cell culture system of infection as a tool in antiviral research, we examined the efficacy of three broad-spectrum antivirals (i.e., FU, FAV, and RBV) against persistent USUV. We have recently demonstrated that USUV is sensitive to all these drugs during a lytic infection [45]. In our previous study, we found reduced USUV titers (and complete viral extinction under certain regimes) in experiments involving five consecutive lytic passages in Vero cells treated with 800 µM FAV, RBV, or FU. These antiviral activities were linked to increased mutagenesis in the populations rescued from treated cells. Here, we have observed that continuous treatment of persistently-infected V and S cells with FU, RBV, or FAV also results in decreased virus titers in the cellular supernatants (Figure 4), with FU and RBV being more effective against USUV than FAV (Table 2). The IC_50_ values obtained for each drug were similar in different cellular contexts but slightly lower for persistently-infected cultures. The selectivity indexes for these compounds were relatively high for all these drugs, except for Vero cells treated with FU which showed elevated toxicity. We decided to investigate if a combination of FAV and RBV (F + R) could lead to improved antiviral activities (Figure 4D,H). We observed generally lower IC_50_ values for an F + R cocktail than for each drug alone suggesting that it could lead to improved efficacies in the treatment of USUV infection (Table 2).

### 3.5. Prolonged Treatment with Nucleoside Drugs Abolishes Infectivity in Persistently-Infected Cell Supernatants

To investigate whether prolonged exposure to a cocktail of F + R could be effective in curing persistently-infected cells, we treated V and S cells for 6 consecutive days in the presence of a concentration of 2000 µM for each drug. We initially tested the antiviral activity of an F + R combination for 6 days on both V and S cells (Appendix A). By day 5 we found undetectable levels of infectious virus in S and V cell supernatants. However, a relapse in viral infectivity was observed when treatment was discontinued (days 7 to 10), suggesting the presence of infectious virus intracellularly and/or at undetectable levels in the supernatant (Appendix A). We also observed a lytic crisis in both cell lines associated with this viral relapse. Lytic crises were never observed in V and S cells during standard passages (Appendix A).

To further investigate the dynamics of virus extinction by antiviral nucleosides in persistently-infected cells, and whether longer exposure to these drugs could lead to completely curing infection, we set up a larger experiment to analyze virus titers and RNA levels at different time points (Figure 5). Owing to the fact that both cell lines show similar susceptibilities to antiviral nucleosides (Table 2, Appendix A) we decided to perform this experiment with S cells only. We also anticipated that since S cells release lower virus titers than V cells at passage 39 (Figure 1C, Figure 4E–H), it might be easier to achieve the complete curing of infectivity. For this assay, we treated S p39 cells for 10 days with FAV, RBV, or F + R at a concentration of 2000 µM for each drug. Low toxicity profiles and precedent studies in vitro and in vivo support the use of such an elevated drug concentration for this assay [51,52,53]. We observed that all three different treatments elicited similar efficacy against USUV, leading to the complete extinction of infectious virus in the cellular supernatant of persistently-infected cells by day 6 (Figure 5B). To certify viral extinction, these supernatants were serially passaged in naïve Vero cells and in the absence of drugs (Figure 5C). This process was repeated three times, using a new monolayer of cells in each passage. With the exception of RBV-treated cells at day 6, no infectivity was recovered in samples collected at days 6−10 for any of these treatment regimes (Figure 5C). This observation supports that prolonged treatments of S p39 with RBV and/or FAV can eliminate persistent USUV in infected cells.

This loss in infectivity was also accompanied by dramatic decreases in the viral RNA levels. By day 10, viral RNA was barely detectable in FAV or RBV-treated cells, and it was completely absent in F + R series, suggesting the complete extinction of USUV in the cellular supernatant (Figure 5C,D). The specific infectivity of USUV was significantly lower in supernatants collected from treated cells than in untreated cells (Figure 5E,F). Decreases in specific infectivity are a typical feature of virus populations approaching extinction as a consequence of lethal mutagenesis. This would be in agreement with our previous observations where we documented an antiviral activity linked to increased USUV mutagenesis during a lytic infection [45].

### 3.6. Interruption of Drug Treatment Results in the Recovery of Viral Infectivity

To examine whether S p39 cells treated for 10 days had been completely cured of the infection, we cultured these monolayers for three additional days in the absence of drugs (Figure 6). We found that, after drug removal, USUV infectivity re-emerged in the supernatant of cells previously treated with RBV or FAV only (Figure 6B). Those cells that had been treated with an F + R cocktail did not show any recovery of infectivity by day 3, even when the supernatant collected was passaged in consecutive blind infections (Figure 6C). However, further culturing of F + R cells in the absence of drugs led to the recovery of infectivity by day 5 (Figure 6D), and complete cytopathology of the monolayer was observed at day 7, as previously observed in experiments described above (Appendix A).

### 3.7. Loss of Extracellular Infectivity is not Accompanied by Intracellular Extinction of USUV

All this evidence suggested that, despite the complete extinction of infectivity in the supernatants of treated cells (days 6–10), infectious viral genomes may be replicating intracellularly and escaping treatment. To investigate this possibility we analyzed the presence of viral RNA in whole cell RNA extracts isolated at different time points (Figure 5A). We confirmed that, despite the rapid extinction of infectious particles and viral RNA in the extracellular environment (Figure 5B–D), there were relatively large intracellular levels of viral RNA detected (Figure 7). By treatment day 10, the amount of viral RNA in cells was only 1.5-log_10_ lower than in mock-treated cells, in striking contrast with the large decreases in virus titers and RNA levels observed in the supernatant of the same cells (~5-log_10_ decrease). To rule out that this disparity may be due to a distinct sensitivity to drugs in standard and defective genomes, we used both qPCR techniques abovementioned to detect either Env- or NS5-coding regions. We confirmed that the intracellular decrease of standard full-length viral genomes was similar to that observed for total viral RNA, which is indicative of a similar susceptibility to drugs in both truncated and standard genomes (compare Figure 7A,B). Thus, our data suggests that the recovery of infectivity after drug withdrawal is due to the replication of competent intracellular viral genomes that have escaped the antiviral treatment.

## 4. Discussion

As a consequence of diverse socio-economic and environmental changes, flaviviruses transmitted by arthropods are a growing source of public health concern. In the last few years, new emerging pathogens causing human disorders with an overall increase in the incidence of flaviviral disease have been documented [11,44,54,55]. Thus, the development of novel cell culture models can be instrumental to further characterize flaviviral infection and to identify more effective antiviral therapies. In this study we have demonstrated that USUV, an emerging zoonotic threat to humans, can establish a persistent infection in Vero cells where it replicates robustly for at least 80 days (Figure 1). V and S cells exhibited normal growth, modestly slower than Vero, leading to confluent monolayers of similar appearances (Appendix A). However, V and S cultures showed larger contact inhibition, lower tendency to grow in multiple layers, and contained larger numbers of round cells detached from the monolayers (Appendix A). Nonetheless, no lytic crises were reported during the whole passage history and extended culture times (e.g., 120 h) did not lead to any increase in the number of detached cells (Appendix A).

The use of cell culture systems has been proposed as a conceivable surrogate approach to characterize the determinants that regulate a persistent infection in the host. In particular, cell culture systems of infection have been developed for HCV, a pathogen genetically related to flaviviruses transmitted by vectors [31,32]. HCV is a major global cause of viral chronic disease with estimates suggesting that 70 million people are currently infected [56]. Although these cell culture systems are not aimed at replicating the complex networks implicated in the establishment of chronic infection in vivo, they can assist in identifying the cellular genes that are distinctly regulated during persistence, and developing more effective therapeutics required during prolonged treatments [28,32]. While there is yet no evidence of USUV establishing persistence in vivo, the fact that subclinical infections occur in birds and humans underpins this possibility [35,37,38]. For closely related WNV, a pathogen causing significant numbers of cases of chronic disease in humans, it has been found that most natural infections in its mammalian hosts (humans, horses) are subclinical [57]. To the present date, there are limited studies in vivo with USUV which hinders the possible development of animal models of persistence [46,58,59]. USUV causes an asymptomatic infection in wild type mice or chickens which encourages using these animal models to develop systems of study persistence in vivo [58,59]. Alternatively, in some bird species USUV develops an asymptomatic or mild infection which could therefore be exploited as plausible animal models of persistence.

The use of cell culture lines can help identify those host genes that are distinctly expressed during persistence relative to the acute phase of a viral infection. This information can be relevant to unraveling the networks activated or inhibited in cells surviving infection. The identification of these host factors can be exploited in the design of antiviral therapies based on drugs targeting them. Evidence supporting the use of in vitro models in the characterization of the cellular and viral determinants modulating persistence has been obtained for TBEV [28]. These precedents stimulate prospective studies to identify the determinants modulating USUV persistence in S and V cells. To this aim, we could make use of different approaches such as comparative transcriptomics or whole cell proteomics of persistently- and lytically-infected cells. For TBEV, a comparative transcriptomics profile revealed that prosurvival oncogenes, such as *AKT2* and *ERBB2*, were overexpressed in persistently infected cells while proapoptotic genes, such as *Bad* and beta interferon-1, were downregulated [28].

Despite the lack of knowledge regarding the mechanism by which USUV establishes persistence, we anticipate that defective viral genomes (DVGs), containing truncations of variable length at the 5’ end, may be playing a major role in this process, in agreement with several lines of evidence [6,7]. All the truncated genomes described here are predicted to encode a polyprotein lacking Env, most of NS1, and partially PrM/M (Figure 2). This is suggestive of these structural genes, and possibly NS1, playing a major role in triggering cell death. A connection between the cytopathic effect observed in flavivirus-infected cells and the expression of Env has been established [6]. The emergence of DVGs lacking or containing truncated versions of Env and NS1 during persistence has been described for other flaviviruses [6,27]. In particular, in TBEV-persistently infected cells, genomes have been identified that lack almost the entire Env-coding region and the N-terminal domain of NS1 [27]. There have also been found deletions in Env and NS5 proteins of dengue virus-2 persistently infecting mosquito cells [60]. In Vero cells persistently infected with Murray Valley encephalitis virus the presence of genomes encoding a truncated NS1 have been documented [61]. In a recent study, WNV genomes isolated from infected birds revealed the presence of truncated genomes containing large deletions, leading to the complete loss of Env and portions of NS1 and PrM/M, similar to what we have observed in USUV [62].

Indirect evidence from nucleoside drug experiments described above underpins the idea of DVGs having a protective role for the host cell. We found that antiviral treatment followed by drug removal generally led to increased cell death associated with the relapse in infectivity (Appendix A and Figure 6D). We posit that decreased intracellular viral RNA levels (Figure 7), which include DVGs, leave the cells ‘unprotected’ against re-infection by fully infectious virus. Further investigations are needed to clarify whether DVGs actually interfere with full-length USUV genome replication, and contribute to cell survival and persistence. There are at least three hypotheses to explain the contribution of defective genomes to persistence. A first model to explain the role of DVGs during persistence sustains that DI particles emerge during standard virus replication and, when they become dominant in the infected cell, they interfere with standard viral genome replication, further decreasing its presence in the cells [6,7]. A second hypothesis based on recent data for Sendai virus posits that DVGs stimulate pathways of cell survival while cells containing elevated amounts of standard genomes follow apoptotic routes [7]. A third model supports that DVGs protect cells from death by a mechanism known as superinfection exclusion, consisting of the ability of an established virus infection to prevent subsequent infections in the same cell. Superinfection exclusion has been described in BHK-21 cells harboring a subgenomic WNV replicon (synthetically produced) which exhibited lower susceptibility to virus infection than naïve cells [63]. WNV superinfection exclusion could be reversed by treating the replicon-bearing cells with a virus inhibitor [63], which is in agreement with increased cytopathology observed in USUV persistently-infected cells after treatment with antiviral drugs.

We hypothesize that this cell culture system for persistent USUV can be instrumental in the identification of effective therapeutics in the control of infection. In recent years, USUV has caused a modest but growing number of cases of human disease, mainly affecting European countries [38,43,44,64]. Several lines of evidence are also hinting at a possible underestimation of the incidence of human disease, with several cases of USUV misdiagnosed as WNV [37,39]. Here, we observe the complete elimination of infectious virus in the extracellular environment of cultures treated with broad-range antiviral nucleosides FAV and RBV at a concentration of 2000 µM. However, relatively large viral RNA levels remained detected intracellularly. While drug concentrations used in our cell culture system are elevated, several lines of evidence suggest that for FAV these levels can be reached in vivo [53,65]. In a non-human primate model of Ebola/Marburg virus infection, treatment with 150 mg/Kg/day FAV led to plasma concentrations of ~1500–2000 µM one hour after dosing [53]. Our results suggest that intracellular viral genomes (both truncated and standard) resist the antiviral activity elicited by these nucleosides and, when drug treatment terminates, they replicate normally and rescue infectivity. We envisage future experiments to further investigate the disparity observed between the intracellular and extracellular environments. In particular, we deem of utmost interest identification of the virus life cycle steps inhibited by these nucleosides (e.g., protein translation, encapsidation, viral egress) which could be accountable for the absence of infectious particles outside the cell. A tentative explanation is that a vast majority of USUV intracellular genomes are replicative-defective RNA molecules as a consequence of an excessive accumulation of mutations in the presence of FAV and/or RBV. Supporting this hypothesis, we have observed significant decreases in the specific infectivity of viral genomes in the cellular supernatants of cells treated for 2 to 4 days (Figure 5F,G).

We have previously examined the efficacy of these drugs against USUV during lytic infections, and established an association between the antiviral activities observed and increased virus mutation frequencies [45]. The use of nucleoside drugs eliciting increased virus mutation rates is currently being investigated as an alternative antiviral strategy known as lethal mutagenesis [52,66,67]. An advantage of therapies based on mutagenic nucleosides, such as those used in this study, is that the antiviral effect associated with mutagenesis (i.e., detrimental mutations) is accumulated during viral RNA synthesis, and thus can be propagated to offspring genomes. In theory, mutagenic drugs should be more effective against persistent infections since longer periods of infection entail higher number of replication cycles, and hence larger number of mutations accumulated. Several lines of evidence from clinical and in vivo studies endorse this hypothesis. We have previously demonstrated that FAV is effective in curing persistent murine norovirus infection in mice, and recent data further support FAV efficacy against chronic norovirus in humans [68,69]. The therapeutic potential of FAV has been recently demonstrated in a mouse model of lethal USUV infection which encourages additional studies to confirm whether this activity is linked to mutagenesis in vivo [46]. Likewise, RBV has been broadly used in the clinical treatment of patients chronically infected with HCV [70]. Notably, in cell culture systems of HCV persistent infection (Huh-7.5 cells) an autophagic response is observed which results in lysosomal degradation of a nucleoside transporter (equilibrative nucleoside transporter 1) utilized by RBV to enter the cell. Persistently infected-cells showed reduced intake of RBV, and hence lower sensitivity to treatment [31]. Additional data confirmed that sensitivity to RBV was regained in HCV-infected cells treated with autophagy inhibitors which restored nucleoside transporter levels in the cellular membrane [31]. It remains to be demonstrated, however, whether flavivirus persistent infection leads to alterations in the levels of nucleoside transporters that could be affecting cellular sensitivity to antiviral drugs. Although our virus titer inhibition data may be suggesting the opposite (USUV is more susceptible to drugs during persistence), these compounds exhibit lower toxicity in persistently-infected cells than in Vero cells, in agreement with this possibility (Figure 4, Table 2).

Full genome sequence analysis of USUV has identified the presence of several non-synonymous mutations, most of them accumulated in the NS4A and NS4B coding regions. Both NS4A and NS4B proteins interact with each other in the infected cell, and the determinants of this interaction have been characterized for dengue virus [71]. The residues in NS4A that participate in direct contact with NS4B include positions 19 to 22 in α-helix 1 (18 to 21 in USUV), and residues A40, A44, and L48 in α-helix 2 (A39, A43, and L47in USUV) [71]. Several amino acid changes identified in our study are in close proximity to residues involved in the interaction with NS4B (Table 1). In particular, mutations E22D and F24L are close to residues 18–21 in helix α1 aforementioned, and E45N is located within residues A39, A43, and L47 in α-helix 2. The mutation T92M in NS4B, which is repeatedly found in both V and S cell lineages, is also located in the NS4B domain predicted to interact with NS4A (Table 1) [71]. The accumulation of these non-synonymous mutations suggests that a strong selective pressure operates on these proteins during USUV persistence. A tantalizing possibility is that these amino acid changes are leading to protein–protein binding alterations which could be favoring persistence in the infected cells.

Flavivirus NS4A and NS4B proteins are essential to the viral life cycle. Both viral products are directly implicated in membrane rearrangements of the endoplasmic reticulum (ER) needed for the formation of the replication complex, and the induction of autophagic pathways that prevent death of the infected cell [72,73,74,75]. Although some studies suggested that flavivirus NS4A alone is sufficient to induce prosurvival PI3K-dependent autophagy, recent findings with ZIKV have revealed that both NS4A and NS4B can cooperatively suppress the Akt-mTOR pathway and trigger autophagy [73,76,77]. For several flaviviruses including USUV, it has been demonstrated that autophagy inhibitors elicit an antiviral activity [75,78]. These findings together with the observations above on the impact of autophagy on cell antiviral sensitivity encourage future investigations on the therapeutic value of drug cocktails including nucleoside analogues and autophagy inhibitors.

We posit that some of those genetic alterations identified in persistent USUV are provoking its attenuation in Vero cells, thus facilitating cell survival to infection. We now aim to investigate whether these changes are leading to USUV infectivity losses not only in mammalian but also in other avian and insect cell lines. These prospective studies can help reveal the viral determinants that modulate virulence in vertebrate and invertebrate hosts.

## 5. Conclusions

The development of cell culture systems based in persistent flavivirus infection can be instrumental to better understanding the intracellular rearrangements implicated in chronic viral disease. Cellular models of persistent infection can also provide better surrogates for in vitro systems to examine the efficacy of antiviral compounds in the treatment of chronic infection. We have validated our model using old-known antivirals which exhibited sustained activity and suppressed detectable viral infectivity in the cellular supernatants during prolonged periods, although virus relapse occurred when drugs were removed. These results encourage the use of this model of infection for the identification of improved therapeutic drugs against the flaviviruses.

## Figures and Tables

**Figure 1 viruses-11-00560-f001:**
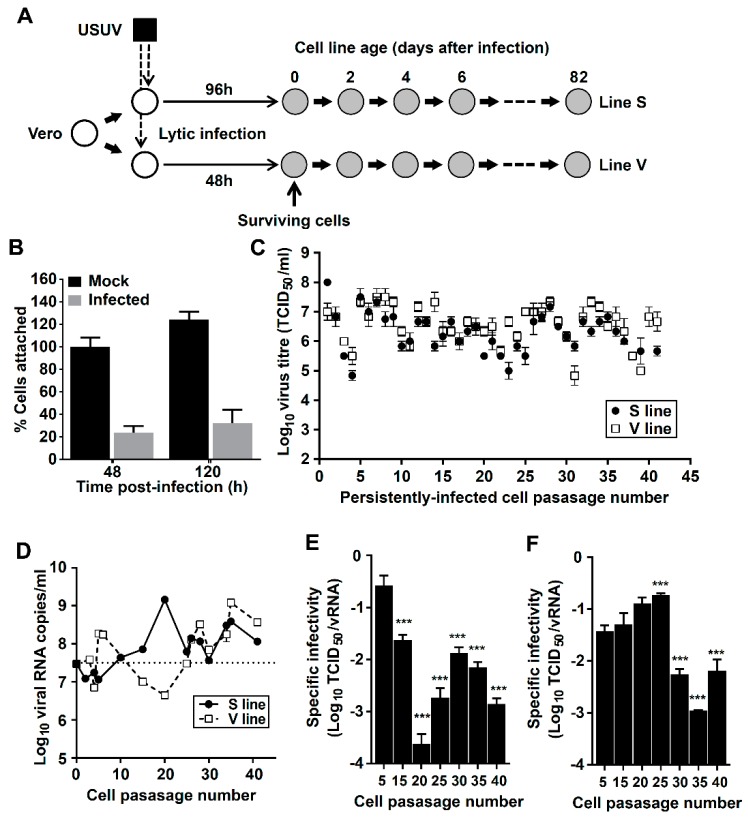
Establishment of cell culture systems of persistent Usutu virus (USUV) infection. (**A**) Schematic representation of the procedure followed to the establishment and maintenance of Vero cells persistently infected with USUV. Uninfected cells are represented as white circles, persistently-infected cells as grey circles, and cell passages with thick arrows. Vero cell monolayers are infected with USUV (black square) at an MOI of 0.01. At 48 (V cells) or 96 h (S cells), unattached dead cells are removed and the surviving cells cultured in fresh media. Cells are passaged every 48 h using standard procedures. (**B**) Percentage of cells that remain attached to a 16-mm dish at 48 or 120 h after USUV (grey) or mock infection (black). These values are relative of mock-infected cells at 48 h post-infection. (**C**) USUV titers in persistently-infected cells along passages. Viral titers in S cells supernatants are represented as black circles, V cells as white squares. (**D**) Viral RNA in cell supernatants of V (white squares) and S cells (black circles) along passages. The number of molecules in each passage was determined by qPCR using primers targeting the Env-coding region. (**E**) and (**F**), USUV specific infectivity in the supernatant of V (**E**) and S cells (**F**). Specific infectivity is determined as the ratio between infectious units (TCID_50_) and the number of viral genome equivalents detected in the samples. Statistically significant differences are represented by asterisks (*** *p* < 0.001). Two-way ANOVA test.

**Figure 2 viruses-11-00560-f002:**
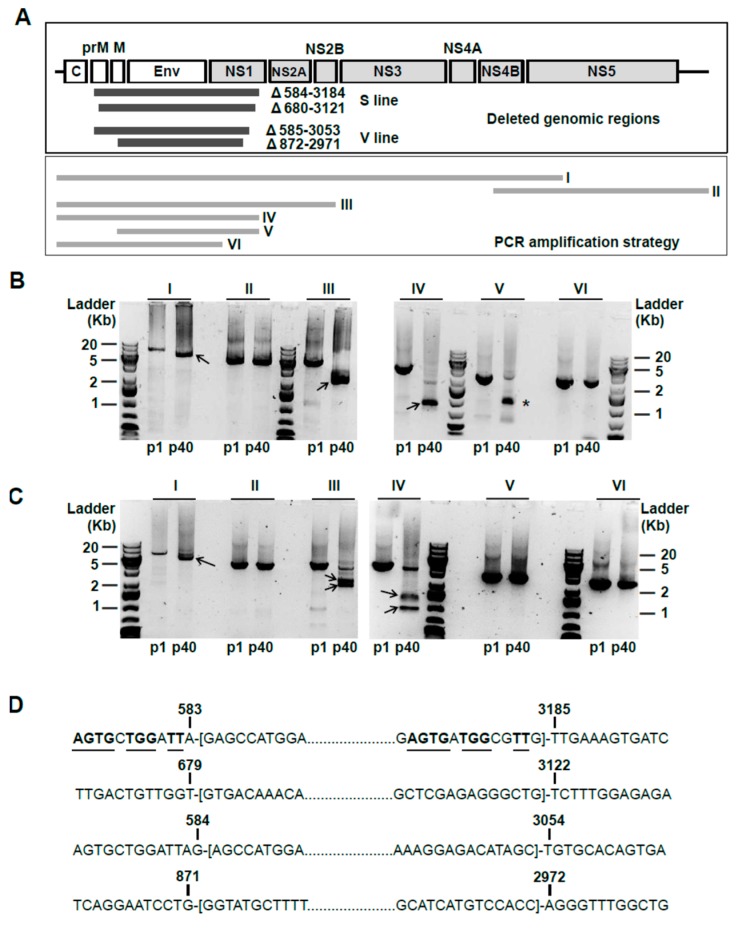
USUV genomes containing large deletions in the 5’ end are detected in persistently-infected cells. (**A**) Schematic representation of USUV genome, PCR amplification strategy, and deletions identified. Top panel, illustration of USUV genome. Each box represents a different viral protein-coding region. White boxes represent genomic sequences encoding structural proteins (C, capsid; M, membrane protein; PrM, M precursor; Env, envelope) and grey boxes depict non-structural proteins (NS1 to NS5). Different deletions are indicated by a thick dark grey line below the corresponding truncated sequence. Bottom panel, PCR amplification strategy (only shown those corresponding to PCR gel analyses below). Each predicted amplicon is represented as a thick light grey line (I to VI). B and C, amplification products of viral RNA extracted from persistently-infected cells at passage 40 (p40) or cells infected with wild type USUV (p1). Amplification of viral RNA from S (**B**) and V cells (**C**) and compared to USUV wild type. The amplification products were run on a 1.5% agarose gel in the presence of 0.5× TBE for 35 min. PCR bands in persistently-infected cells (p40) of lower molecular mass than those found in the standard genome (p1) are indicated with an arrow. Viral RNA from S p40 (**B**) amplified with primers spanning residues 818 to 3359 (amplicon V) revealed the presence of a new deletion product (~1.5-Kb shorter than in wild type virus). This new truncation was not apparent in larger amplification products (amplicons I, III, and IV). (**D**) Detail on USUV genomic sequences surrounding the deletion sites. The deleted sequences are represented between brackets. Nucleotide positions immediately upstream and downstream of the deleted sequence are numbered according to reference USUV sequence AY453411. There is no apparent homology in the genomic regions surrounding the nucleotides at the joining site, except for the first sequence provided, with putative identities underlined.

**Figure 3 viruses-11-00560-f003:**
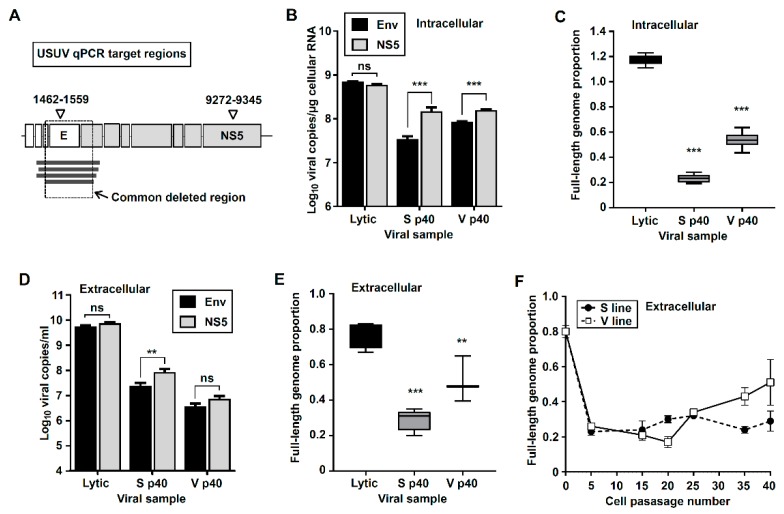
Detection of defective viral genomes in persistently-infected cells. (**A**) Illustration of USUV genome and target sequence sites for different qPCR strategies. Different deletions found in V and S defective genomes are shown as thick dark grey lines, and a square indicates the common deleted region in these different sequences. The procedures for each qPCR detection method are described in Materials and Methods. Methods for the detection of non-structural protein 5 (NS5)-coding region are used for the quantification of total viral RNA molecules (including both standard and truncated viral genomes). Methods for the detection of the Env-coding region are used for the specific detection of standard genomes (excluding defective genomes). (**B**,**C**) Intracellular detection of viral RNA. B, quantification of intracellular viral RNA in lytically- and persistently-infected cells using qPCR detection methods for Env (black bars) or NS5 (grey bars) coding regions. Each bar represents the average of values obtained from independent cell monolayers (*n* > 5). (**C**) Estimated ratio of standard length genomes in the sample (viral RNA molecules detected with Env relative to NS5, based on results shown in (B). (**D**,**E**) Same as in B and C but in samples extracted from cellular supernatants (*n* > 3). (**D**) Quantification of extracellular viral RNA using qPCR detection methods for Env (black bars) or NS5 (grey bars) coding regions. Each bar represents the average of values obtained from independent cell monolayers (*n* > 5). (**E**) Estimated proportion of full-length genomes in the supernatant of V and S cells at p40, and Vero cells infected with wild type USUV. (**F**) Proportion of full-length genomes in the supernatants of S and V cells along passages. Statistically significant differences are represented by asterisks (* *p* < 0.05, ** *p* < 0.01, *** *p* < 0.001). B and D, one-way ANOVA test. C and E, two-way ANOVA test.

**Figure 4 viruses-11-00560-f004:**
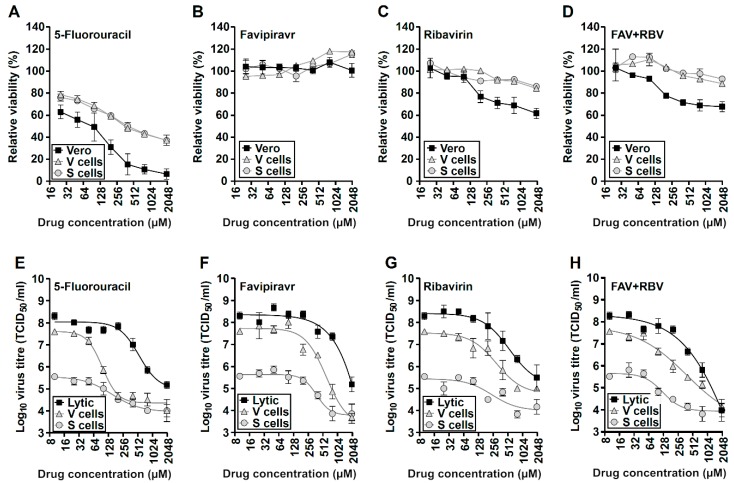
Antiviral nucleosides inhibit USUV persistent infection in Vero cells. A-D, Cellular toxicity displayed by different nucleoside drugs was tested upon V p39 (grey triangles), S p39 (grey circles), and uninfected Vero cells (black squares), using the CellTiter–Blue Cell viability assay (Promega), which accounts for living cells. Cellular viability was measured in cells treated with increasing concentrations of 5-fluorouracil (FU) (**A**), favipiravir (FAV) (**B**), ribavirin (RBV) (**C**), or FAV + RBV (**D**) at 48 h and represented as a relative value to untreated cells (*n* = 3). (**E**,**F**) Viral titers recovered in V p39 (grey triangles), S p39 (grey circles), and lytically-infected Vero cells (black squares), after 48h in the presence of nucleoside drug analogues. Values observed in cells treated with increasing concentrations of FU (**E**), FAV (**F**), RBV (**G**), or FAV + RBV (**H**). To this assay, naïve uninfected Vero cells (black squares) were inoculated with wild type USUV at an MOI of 0.01, and after adsorption the cells were treated with drugs for 48 h. Persistently-infected V p39 and S p39 were directly treated with drugs for 48 h. Every time point is the average of three (*n* =3) biological replicas (±SEM). In FAV + RBV graphs (D,H) the ordinate values represent the concentration for each drug individually (e.g. 400 µM refers to a cocktail of 400 µM FAV and 400 µM RBV, or 800 µM nucleosides).

**Figure 5 viruses-11-00560-f005:**
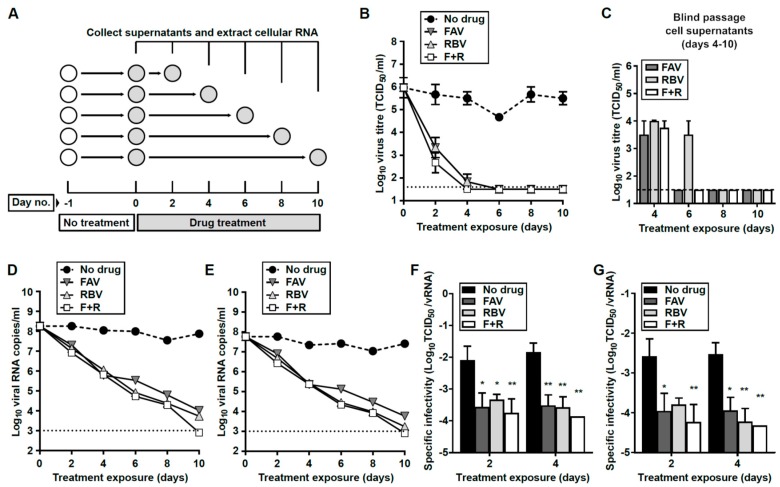
Treatment with FAV, RBV, or a combination of both can cause complete loss of infectivity in cell supernatants. (**A**) Schematic representation of the experimental procedure. For each treatment, six independent monolayers of S p39 cells were seeded (*n* = 6). Whole monolayers (RNA extraction) and supernatants (RNA and virus titers) of untreated or treated cells were collected at days 2, 4, 6, 8, and 10 (*n* = 1). (**B**) Virus infectivity in S p39 cells treated with FAV, RBV, or a cocktail of both drugs (F + R) at 2000 µM each drug. A dotted line represents the limit of detection. (**C**) Virus titers recovered after a blind infection in naïve Vero cells using 100 µL of S p39 cell supernatants obtained in B after treatment. Those samples containing undetectable virus levels (dashed line) remained negative when two additional blind passages in Vero cells were performed (not shown). (**D**,**E**) Viral RNA levels in the supernatants of treated cells, using qPCR methods based on the detection of NS5- (D) and Env-coding regions (**E**). (**F**,**G**) Specific infectivity (SI) of USUV genomes detected in the cellular supernatants of treated cells. SI is calculated as the ratio of infectious units (**B**) to viral RNA molecules (**D** and **E**). SI based on qPCR methods for the detection of NS5 (**F**) or Env (**G**) genes.

**Figure 6 viruses-11-00560-f006:**
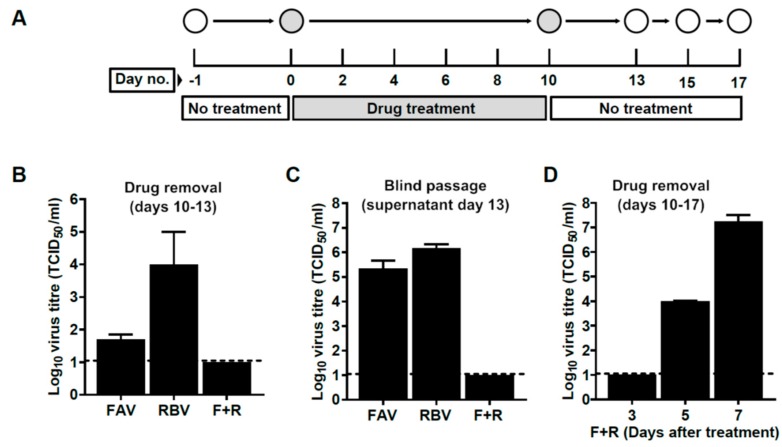
USUV infectivity re-emerges in the supernatant of persistently-infected cells after drug withdrawal. (**A**) Experimental procedure. Cells were treated for 10 days in the presence of FAV, RBV, or a combination of both (F + R) at a concentration of 2000 µM for each drug (*n* = 1). After 10 days, drugs were removed and cells cultured for 3 (FAV and RBV) or 7 additional days (F + R). (**B**) Viral infectivity (TCID_50_/mL) detected in the supernatant of cells after 10-day drug treatment followed by 3 days in the absence of drugs. (**C**) Viral infectivity recovered after three serial passages of cellular supernatants collected in B. In each blind infection, 100 µL of supernatant from the previous passage was used to infect a new monolayer of Vero cells. After adsorption for 1 hour, fresh media was added and cells were incubated for 24 additional hours at 37 °C. (**D**) Viral infectivity (±SEM) in the supernatant of S cells treated with F + R for 10 days followed by absence of treatment for 3, 5, and 7 days. (B–D) A dashed line represents the limit of detection.

**Figure 7 viruses-11-00560-f007:**
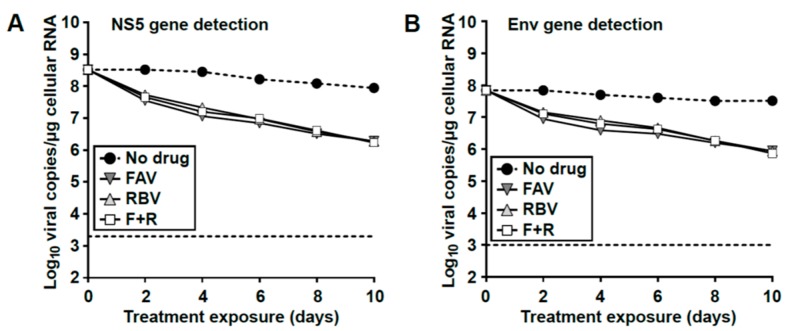
Intracellular viral RNA detected in S p39 persistently-infected cells treated with antiviral drugs. These RNA extracts were obtained from the experiment represented in Figure 5A. Viral RNA levels detected in whole cell monolayer extracts after treatment with FAV, RBV, or a combination of both drugs (F + R) at a concentration of 2000 µM for each drug. For each treatment regime, 6 independent wells were seeded (*n* = 6) and samples were collected at different time points (*n* = 1). Quantitative PCR (qPCR) methods, based on the detection of NS5- (**A**) and Env-coding regions (**B**) are used. A dashed line represents the limit of detection.

**Table 1 viruses-11-00560-t001:** Genomic differences between parental stock (wild type) and persistent USUV (isolated from V p40 and S p40), and compared to the USUV reference strain (GenBank ID: AY453411).

Nucleotide Position	USUV Parental Stock	S p40	V p40	Viral Gene	Amino Acid Replacement
G775 *	A	A	A	M	E9K
C891 ^1^		U	U	M	−
C1815 ^1^		U	U	Env	−
C3661			U	NS2A	−
A3867			G	NS2A	−
C4958			U	NS3	T118M
C5432 *	U	U	U	NS3	A276V
G6528		U		NS4A	E22D
A6532 ^2^		C		NS4A	F24L
U6534 ^2^			A	NS4A	F24L
G6595 ^1^		C	C	NS4A	E45N
U7175		G		NS4B	L89R
C7184 ^1^		U	U	NS4B	T92M
U7197 *	G	G	G	NS4B	F96L
U7288			C	NS4B	Y127H
U7424			G	NS4B	M172R
A10311 *	G	G	G	NS5	−

* An asterisk denotes sequence differences between our USUV stock and the reference strain from which it derives (Genbank ID AY453411). ^1^ Those nucleotide substitutions were found in both V and S persistently-infected cells. ^2^ Two different nucleotide substitutions in V and S cells are leading to the same amino acid replacement.

**Table 2 viruses-11-00560-t002:** Antiviral activity of nucleoside analogues against USUV in lytically and persistently-infected cells.

Cell Line	Compound	IC_50_ (µM)	CC_50_ (µM)	Selectivity Index
**Vero cells**	5-Fluorouracil	71	66	1.1
	Favipiravir	315	>2000	>6.3
	Ribavirin	163	>2000	>12.3
	Favipiravir + Ribavirin	60 *	>2000 *	>33.3

**V p39**	5-Fluorouracil	42	467	11.1
	Favipiravir	197	>2000	>10.2
	Ribavirin	71	>2000	>28.2
	Favipiravir + Ribavirin	36 *	>2000 *	>55.6

**S p39**	5-Fluorouracil	56	472	8.4
	Favipiravir	241	>2000	>8.3
	Ribavirin	80	>2000	>25.0
	Favipiravir + Ribavirin	78 *	>2000 *	>25.6

The 50% cytotoxic concentration (CC_50_) and the 50% inhibitory concentration (IC_50_) were extrapolated from nonlinear regression curves obtained from experiments represented in Figure 4 (*n* = 3). * An asterisk indicates that this value is for each independent drug. Thus the IC_50_ in these samples for both drugs taken into consideration together are 120, 72, and 156 µM, respectively. The CC_50_ values would be all >4000 µM.

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
