# Peer review of "Establishment of a Cell Culture Model of Persistent Flaviviral Infection: Usutu Virus Shows Sustained Replication during Passages and Resistance to Extinction by Antiviral Nucleosides"

_viruses, 2019, doi:10.3390/v11060560_

Round 1

Reviewer 1 Report

Manuscript by Navarro and Arias (viruses-479319) report the establishment of a cell culture model for USUV persistence and its usefulness to test antivirals.

Even though, a quite huge amount of experiments have been conducted, I have serious concerns about the interpretation of some of the data.

Some paragraphs are confusing and quite difficult to follow

To establish persistence in the cells, the authors claim that at 48h post-infection around 80% detached from the plates. This is quite unusual, as flaviviruses, and specifically USUV, use to take longer time to produce extensive cytopathic effect, mainly when using a low moi, as it is the case (0.01). They claim that cells remained attached to the plates even at longer post-infection times, but no data are shown. Why did they chose to propagate the supposed persistently infected cells every 48h? How can they be sure that no cytopathic effect will be observed at longer times? With this schedule, and refreshing the media in each passage, it is quite possible that they are seeing new lytic infections of not infected cells. In fact, for viral titers determination (line 142) they score the number of infected cells 5 d.p.i.

No explanation about the differences observed between V and S lines is included.

The concentration of drugs used are too high, far away to what it can be envisage for clinical practice, thus reducing the relevance of the work. Cell viability should be shown, mainly with such a high concentrations, as well as the EC50, IC50 and SI.

Specific points

Number of replicates should be indicated for all the experiments, and not just for some of them.

To avoid unnecessary reiterations, Figure legends should not include a description of the results

In Fig 1 it is indicated that the specific infectivity (TCID50/viral genome equivalents) at the initial passage is 0, but this is incorrect, as there always be more viral RNA than infective virions. In fact, in fig 7F-G it is not cero.

Fig 1 E-F shows similar levels of viral RNA for both E and NS5 genes in both cellular lines, should not be lower for E gene due to the detected deletions?

Calculations of the log10 of the viral RNA copies/ml is missing in material and methods, which otherwise are quite detailed.

How many cell passages has the used virus? This can be important, as the virus could have adapted to the cell line used, which is not a natural target of USUV. In fact, it is stated that there are differences between the original strain and the derived one used.

From one of the primers set (2976/6992) no results are shown.

Line 101. “…in each cell”, should be “…in each cell line”

Lines 256-259. Are they talking about fragment III? IV? Please, clarify.

Lines 261-263. What about fragment I that is slightly shorter than the full genome?

Line 303. How early? If very early it can be lytic infection.

Line 303. If defective particles are being release to the supernatants, why there is not a reduction of infective viral titer at passages 10, 30 and 35 (Fig 1C)

Line 330 and Fig 4 B-C. Why passage 9 was used instead of passage 40 as in other experiments? And the same for Fig 6.

Fig 5. What about V cell line with 2000um?

Line 378-379. Could not be that by passaging every two days there were no time to observe a possible lytic crisis?

Line 381. Why only line S, and not V, was treated during 10 days?

Line 385. For how long were the infected supernatant leaved on the monolayers? Was this enough time to asses it? In fact, at the end, virus recovered is reported (Fig 8).

Line 436-437. The statement that “there was a rapid extinction of infectivity…..” is contradictory to the detection of viral RNA at days 6, 8 and 10 (lines 439-441)

Line 471-473. How do they propose to use the system to identify cellular determinants of persistence?

Line 526-527. With the experiments conducted is quite difficult to suggest that the virus has different sensitivity to drug treatment during persistence than during lytic infections, as the drug were added only once and keep for several days, instead of adding fresh drug in sequential days.

Line 566-568. As infectivity is finally recovered, this sentence does not seem appropriate.

References should be checked it, as some time they are not correctly quoted.

Author Response

Dear reviewer #1,

We appreciate the time dedicated to review our manuscript and your comments made on the text. We deem your suggestions very valuable and consider that our revised version is now a much more improved paper than our previous submission to Viruses. We have now introduced extensive modifications to make our data clearer and to respond all your queries. In particular, we have carried out IC50, CC50 and selectivity index determinations which are now included in Figure 4 and Table 2. This information has permitted us to clarify that USUV in persistence is slightly more sensitive to nucleoside drugs than in lytic infections, in contrast with previous observations. This fact doesn’t affect the main message of our manuscript but we have now introduced some modifications to clarify these new observations.

Besides, in the revised version we have introduced the following changes:

-  Format modifications of some panels of Figures 1 and Figures 3

-  Figure 4 and 5 have been deleted and replaced with new data (Figure 4 and Table 2). These data include IC50, CC50 and selectivity index determinations

-  Figure 6 has been moved to Supplementary data (Figure S3)

-  New data has been included as supplementary information (Figures S1 and S2)

- Modification of the title to make it more accurate

Specific responses to specific comments are provided below

Reviewer 1

R: To establish persistence in the cells, the authors claim that at 48h post-infection around 80% detached from the plates. This is quite unusual, as flaviviruses, and specifically USUV, use to take longer time to produce extensive cytopathic effect, mainly when using a low moi, as it is the case (0.01). They claim that cells remained attached to the plates even at longer post-infection times, but no data are shown.

We agree with the reviewer that some additional information on this should be provided. Now we have included new data in Figure 1B. There are always 20-25% cells that remain attached to the plate even at late time points (e.g. 5 days post infection)

R: Why did they chose to propagate the supposed persistently infected cells every 48h?

In our laboratory we regularly pass stock Vero cells three times a week for maintenance (two days at 48h and one day at 72 h, weekend). Based on our regular regime for cell passages we decided to pass persistently infected cells every 48 hours.

R: How can they be sure that no cytopathic effect will be observed at longer times? With this schedule, and refreshing the media in each passage, it is quite possible that they are seeing new lytic infections of not infected cells. In fact, for viral titers determination (line 142) they score the number of infected cells 5 d.p.i.

We agree with the reviewer that we have provided insufficient information on the properties of the cells during persistence. We have now included a supplementary figure (Figure S1) where we clearly show that persistently infected monolayers can be maintained for at least 5 days without passaging. We have also seen that both V and S monolayers can be maintained indefinitely (>16 days) if media is replaced when is getting yellow (without need for trypsinisation). This is similar to what is observed for regular Vero cells.

We have also included modifications in the text where we describe alterations in cell properties: slightly slower growth, larger contact inhibition, and a larger proportion of round cells but never leading to extensive cytopathology of the monolayer, e.g. compare similar aspect in V p39 cells at days 3 and day 5 after seeding (Figure S1). The text has been modified in Lines 303-308 in marked-up manuscript.

Regarding titrations: Our Usutu virus strain replicates fast in Vero cells. At high MOI, extensive cytopathology is observed at 24 hpi, at MOI 0.01 extensive cytopathology is observed at 48 hpi. We generally score USUV titrations at 4 days although the results are normally indistinguishable from observations at day 3. It is important to have in mind that the largest possible dilution (1 TCID50 in 10,000-20,000 cells) equals an MOI <0.0001. I deem is relevant to mention that different strains (e.g. persistent virus) or debilitated virus (e.g. after mutagenesis) can take longer than our reference USUV stock, and thus 4 or 5 days can be relevant in certain specific cases to score titrations.

Q: No explanation about the differences observed between V and S lines is included.

The main goal of including two lines of persistent infection is to add robustness to our data. This has been especially relevant to interpret several results observed here. We found that V and S both show fluctuating titres along passages but without tendency to increases or decreases in titres (Figure 1C), both show a tendency to reduced specific infectivity (Figure 1E and 1F) and both viruses lead to similar truncations and the accumulation of 4 identical mutations. This reproducibility in two independent lineages supports that these are biologically relevant features to the establishment of persistence in cells. We have now included one sentence in lines 116-118 to clarify this.

Q: The concentration of drugs used are too high, far away to what it can be envisage for clinical practice, thus reducing the relevance of the work. Cell viability should be shown, mainly with such a high concentrations, as well as the EC50, IC50 and SI.

We have now included data showing IC50, CC50 and selectivity index values (Figure 4, Table 2). We have also modified the text accordingly (Lines 188-218, 823-866, 872-885). We agree that the concentrations used in the assays are high, but we have now included references to in vivo studies where is shown that administration of favipiravir to non-human primates leads to similar concentrations in plasma than those used in the assay (1500-2000µM). Lines 1088-1089, 1347-1351.

We also argue that favipiravir antiviral activity against Usutu virus has been already demonstrated in vivo (Segura Guerrero et al). Lines 131-133, 1390-1392. This supports the relevance of our data in vitro, using this cell culture method as a reference standard to identify antivirals with larger efficacy than favipiravir (e.g. completely curing infection?).

Specific points

Q: Number of replicates should be indicated for all the experiments, and not just for some of them.

This has been amended

Q: To avoid unnecessary reiterations, Figure legends should not include a description of the results

We agree. We have shortened some figure legends accordingly

Q: In Fig 1 it is indicated that the specific infectivity (TCID50/viral genome equivalents) at the initial passage is 0, but this is incorrect, as there always be more viral RNA than infective virions. In fact, in fig 7F-G it is not cero.

We have now modified this Figure (as suggested by another referee) and removed day 0 from the graph. We were using a lytic infection to calculate this number and we believe that we only should include persistently infected cells data.

Q: Fig 1 E-F shows similar levels of viral RNA for both E and NS5 genes in both cellular lines, should not be lower for E gene due to the detected deletions?

The reviewer is absolutely correct. We are totally embarrassed we made such a gross mistake in preparing this Figure. As there are obvious differences in values obtained with qPCRs for E and NS5 we have decided to omit the information for NS5 here. We consider that we should not anticipate this information to the data where we show the presence of truncated genomes (Figures 2 and 3). The new graph showing these differences is now included as Supplementary Material (Figure S2).  

Q: Calculations of the log10 of the viral RNA copies/ml is missing in material and methods, which otherwise are quite detailed.

We agree. We have now expanded the corresponding section in Methods. Lines 279-289 in marked-up version.

Q: How many cell passages has the used virus? This can be important, as the virus could have adapted to the cell line used, which is not a natural target of USUV. In fact, it is stated that there are differences between the original strain and the derived one used.

The virus used in this study has been previously passaged 7 times in Vero cells. This information is now provided in lines 163-165 of marked-up version.

Q: From one of the primers set (2976/6992) no results are shown.

Firstly, we made a minor mistake in the text. Amplifications were 1/7950 and 2976/7950. We have now corrected this.

Second, these amplicons were not included in the Figure due to economy of space purposes. For each virus, we used 2 gel lanes. Each lane contained 10 wells: 2 markers, 6 samples and 2 blanks. For clarity, we never loaded PCRs of two different genomic regions in consecutive lanes. We only included the two amplicons that cover the entire genome, and 4 additional amplicons that help to define the deleted region.

The 2976/7950 amplicon did not provide with additionally relevant information (WT indistinguishable from persistent virus). The information on this PCR is nevertheless included as it was used for sequencing.

Line 101. “…in each cell”, should be “…in each cell line”

Corrected

Lines 256-259. Are they talking about fragment III? IV? Please, clarify.

Amended

Lines 261-263. What about fragment I that is slightly shorter than the full genome?

To maintain clarity in this section (which is quite dense) we have decided to keep this information in the Figure legend and not the main text. 

Line 303. How early? If very early it can be lytic infection.

Here, we refer to an early passage not to an early time point after seeding. We have now amended the text in lines 753-755 to make this information clearer.

Line 303. If defective particles are being release to the supernatants, why there is not a reduction of infective viral titer at passages 10, 30 and 35 (Fig 1C)

It is difficult to extrapolate how the shedding of extracellular defective particles is affecting to virus titres. We observe large fluctuations in titres along passages which could be accounting for alterations in the intracellular balances between defective and standard genomes.

Nevertheless, the proportion of defective genomes in the sample seems to range 50-75% of the total genomes (Figure 3). In the logarithm scale, a 50-75% drop corresponds to a decrease of 0.3 to 0.6 units which may not be appreciated when fluctuations in titres are ranging 2-3 units. Our current hypothesis is that coevolution dynamics between cells getting more resistant to infection and virus becoming more infectious could be also responsible for these fluctuations observed in virus titres (see Herrera et al J Gen Virol. 2008 Jan;89(Pt 1):232-44). Thus, these fluctuations in virus titres might be multifactorial.

Q:Line 330 and Fig 4 B-C. Why passage 9 was used instead of passage 40 as in other experiments? And the same for Fig 6.

We agree, this is a rookie mistake. The new data provided in the Figures of the main text are all including passage 39 populations. Figure 6 (work with passage 35) has now been moved to Supplementary data.

Q: Fig 5. What about V cell line with 2000um?

The new data provided in Figure 4 include treatment of V cells with 2000 uM drugs

Q: Line 378-379. Could not be that by passaging every two days there were no time to observe a possible lytic crisis?

Please see response above and new data provided in Figure S1 (cell monolayers remain unaltered during 5 days). Also consider, that cell monolayers can be maintained indefinitely (>16 days) if media is replaced when getting yellowish.

Although we understand the reviewer concerns, we consider this possibility extremely unlikely. As mentioned before, during a standard lytic infection at high MOI (persistently infected cells are always in the presence of extracellular virus of ~MOI of 1) Vero cells show extensive cytopathology at 24 h pi. In the unlikely scenario that infected cells escape cytopathology in the first passage, they should develop cytopathology when seeded in fresh media as they are already infected.

Q: Line 381. Why only line S, and not V, was treated during 10 days?

We carried out a preliminary experiment where both cell lines were (apparently) cured in the presence of favipiravir and ribavirin (old Figure 6, now Figure S3). Owing to this similar sensitivity to treatment, we decided to focus on a single cell line to characterise the molecular dynamics of extinction. This experiment entailed a complex experimental set up, and thus we decided to work only with one cell lineage. We have now added further clarification in the main text (lines 1081-1088).

Q: Line 385. For how long were the infected supernatant leaved on the monolayers? Was this enough time to asses it? In fact, at the end, virus recovered is reported (Fig 8).

We performed three consecutive 25-hour infections (1 hour adsorption) and 24 hours replication. We have now added further clarification in lines 1151-1152. Our USUV strain leads to large virus titres detected in the supernatant as early as 12 hpi (Bassi et al 2018, Antimicrob Agents Chemother. 2018 Aug 27;62(9).). Our standard passages of lytic infection are also normally performed in 24 hours (extensive cytopathology). This amount of time should be enough for allowing replication of any infectious material, and after three cycles should be leading to high titres of infectious virus (see titres recovered in Figure 8C for FAV, despite almost undetectable levels in 8B).

We consider that the infectivity recovered is a consequence of intracellular viable viral genomes that are not eliminated during treatment (Fig. 9), and thus are capable of replicate and restore infectivity when drugs are withdrawn. Although infectivity outside the cells is completely eliminated, viable genomes remain inside the cells. We have now modified the text to make this clearer. Lines 1178, 1186-1188

Q: Line 436-437. The statement that “there was a rapid extinction of infectivity…..” is contradictory to the detection of viral RNA at days 6, 8 and 10 (lines 439-441)

We apologise for not having made this information clearer. We have now slightly modified this sentence to make it more factual. As referred above, our data indicate absence of infectious virus outside the cell but presence of viral genomes inside the cells that may be responsible for this recovery of infectivity.

Q: Line 471-473. How do they propose to use the system to identify cellular determinants of persistence?

We have now included some additional sentences to clarify what possible strategies can be followed and also to include data with other viruses. Lines 1243-1250

Q: Line 526-527. With the experiments conducted is quite difficult to suggest that the virus has different sensitivity to drug treatment during persistence than during lytic infections, as the drug were added only once and keep for several days, instead of adding fresh drug in sequential days.

We totally agree with the reviewer. New data obtained in Figure 4 following the reviewer indications, suggest that we were wrong. We have modified this part of the discussion accordingly. Lines 1399-1404

Q: Line 566-568. As infectivity is finally recovered, this sentence does not seem appropriate.

We have amended this sentence to make it more accurate and highlight that virus is recovered in the absence of drugs. Lines 1493-1494

Q: References should be checked it, as some time they are not correctly quoted.

We appreciate your suggestions. We have checked and identify some references misplaced

Reviewer 2 Report

The paper by Sempere and Arias describes experiments that led to the establishment of Vero cells persistently infected with the mosquito-borne Usutu virus (USUV). USUV persistence was established following a lytic crisis which ensued after only 48 hours post infection. Persistently infected cell cultures were passaged for 40 times and the authors observed genome truncations in some of the viral genomic RNA. They further tested nucleoside analogs to determinie the effect of these on USUV persistence. The study is an interesting one and a relevant one considering the emergence of vector-borne flavviruses, such as Zika virus and Powassan virus which cause devastating infections in humans. The paper is generally well written, but it needs revisions of English language usage.

Specific comments

Line 37: significant associated fatalities

Line 41: ...infections, such as...

Line 44: ...documented that....

Line 47: Please cite Murray et al, Am J Trop Med Hyg. 2017 Dec;97(6):1913-1919 as well

Line 59: for long periods

Line 60: The life of ticks can be several years: Consider references on TBEV persistence in tick cell lines as well e.g., Mlera et al, Viruses. 2016 Sep 10;8(9). pii: E252; Offerdahl et al PLoS One. 2012;7(10):e47912.

Line 94: Does the historic misdiagnosis suggest that there could be USUV persistence in humans because WNV is known to persist in humans very well?

Line 111: in this study, we found...

Line 117: models for...

Line 127: with 5% CO2.

Line 138: To analyze...

Line 146: To examine the effect off...cells, we seeded...

Line 148: 200-400; how does these high compound concentrations translate to what can be usable in clinical practice?

Line 156: To amplify the USUV...

Line 171: To detect total viral...

Line 187: To establish persistently...

Line 190: Could PBS be used to wash as well?

Figure 2: The authors need to present the truncated genomes in supernatants to show that the truncated genomes were being packaged into particles.

Section 3.3: can the authors suggest how these mutations or specific viral proteins containing the mutations could be contributing towards viral persistence? e.g. what could be the role of NS4A or NS4B in viral persistence?

Section 3.4: It is quite puzzling that the antivirals work very well for the wild-type USUV, but do not work as well against the portion of the wild-type virus in the background of truncated genomes until higher doses are used.  The expectation would be that there are fewer wild-type genomes and the truncated forms cannot replicate efficiently without the help of the wild-type virus. What could be going on here?

Line 381: cells for...

Line 372: We initially...

Section 3.6: It is also puzzling that withdrawal of treatment results in rescue of USUV when it appeared that there were no more infectious particles. Could the authors stretch the treatment time beyond what is presented or they expect no improvement when the drugs are withdrawn?

Line 423: ...monolayers for...

Figure 4 and 5: Proliferation/toxicity assays should accompany these figures to indicate the level of toxicity associated with the compounds.

Line 373: We initially...

Line 381: for 10 days...

Line 413: cells were treated...

Line 423: monolayers for...

Line 466: Are some bird species more susceptible than others and is there a possibility of developing an animal model of USUV persistence. The failure to completely destroy viral RNA following removal of drugs may require involvement of the adapative immune system.

Line 474: Despite the lack of knowledge regarding...

Line 499: Can persistently infected Vero cells superinfected?

Line 505: mainly affecting European...

Line 567: but the virus relapses when drugs are withdrawn!!!

Author Response

Dear reviewer #2,

We appreciate the time dedicated to review our manuscript and your comments made on the text. We deem your suggestions very valuable to prepare a modified version of the paper. We consider our revised manuscript much improved respect to our first submission. We have introduced extensive modifications to make our data clearer and to respond all your queries and those from other reviewers. In particular, major changes include new experiments with antiviral drugs now using the same stock of cells for all the experiments which are now included in Figure 4 and Table 2. This information has permitted us to clarify that USUV in persistence is slightly more sensitive to nucleoside drugs than in lytic infections, in contrast with previous observations. This fact doesn’t affect the main message of our manuscript but we have now introduced some modifications to clarify these new observations.

Besides, in the revised version we have also included the following changes:

-  Format modifications of some panels in Figures 1 and Figures 3

-  Figure 4 and 5 have been deleted and replaced with new data (Figure 4 and Table 2). These data include IC50, CC50 and selectivity index determinations

-  Figure 6 has been moved to Supplementary data (Figure S3)

-  New data has been included as supplementary information (Figures S1 and S2)

- Modification of the title to make it more accurate

Specific responses to specific comments are provided below

Reviewer 2

The paper by Sempere and Arias describes experiments that led to the establishment of Vero cells persistently infected with the mosquito-borne Usutu virus (USUV). USUV persistence was established following a lytic crisis which ensued after only 48 hours post infection. Persistently infected cell cultures were passaged for 40 times and the authors observed genome truncations in some of the viral genomic RNA. They further tested nucleoside analogs to determinie the effect of these on USUV persistence. The study is an interesting one and a relevant one considering the emergence of vector-borne flavviruses, such as Zika virus and Powassan virus which cause devastating infections in humans. The paper is generally well written, but it needs revisions of English language usage.

We have substantially changed the text in this new version to include data and suggestions made by the reviewers. These modifications include a thorough revision of the text and editing those sentences that didn’t sound correct.

Specific comments

Line 37: significant associated fatalities

Amended

Line 41: ...infections, such as...

Corrected

Line 44: ...documented that....

We have corrected this

Line 47: Please cite Murray et al, Am J Trop Med Hyg. 2017 Dec;97(6):1913-1919 as well

We agree. This reference perfectly fits here

Line 59: for long periods

This has been amended

Line 60: The life of ticks can be several years: Consider references on TBEV persistence in tick cell lines as well e.g., Mlera et al, Viruses. 2016 Sep 10;8(9). pii: E252; Offerdahl et al PLoS One. 2012;7(10):e47912.

We have modified the text and included these two references here as suggested

Line 94: Does the historic misdiagnosis suggest that there could be USUV persistence in humans because WNV is known to persist in humans very well?

We have now edited this sentence to make this possible connection more evident to the reader. Lines 111-114

Line 111: in this study, we found...

Amended

Line 117: models for...

Amended

Line 127: with 5% CO2.

We agree. This has been corrected

Line 138: To analyze...

Agree

Line 146: To examine the effect off...cells, we seeded...

Agree

Q: Line 148: 200-400; how does these high compound concentrations translate to what can be usable in clinical practice?

We have now included data showing IC50, CC50 and selectivity index values for all the drugs tested (Figure 4, Table 2). Even though, we agree that these concentrations are extremely high, we have now included recent evidence in vivo where is demonstrated that the administration of favipiravir to non-human primates can leads to similar concentrations in plasma than those used in the assay (1500-2000µM). Lines 1088-1089, 1347-1351.

We also argue that favipiravir antiviral activity against Usutu virus has been already demonstrated in vivo (Segura Guerrero et al). Lines 131-133, 1390-1392. This supports the relevance of our data in vitro, using this cell culture method as a reference standard to identify antivirals with larger efficacy than favipiravir (e.g. drugs that could achieve completely curing infection?).  

Line 156: To amplify the USUV...

Checked and corrected

Line 171: To detect total viral...

Amended

Line 187: To establish persistently...

Amended

Line 190: Could PBS be used to wash as well?

Yes, indeed. We normally use DMEM but in certain occasions we have employed PBS. We have edited the sentence to include this.

Q: Figure 2: The authors need to present the truncated genomes in supernatants to show that the truncated genomes were being packaged into particles.

We have included new data in Figure S2 where we show the same truncations in the supernatants than those found intracellularly

Q: Section 3.3: can the authors suggest how these mutations or specific viral proteins containing the mutations could be contributing towards viral persistence? e.g. what could be the role of NS4A or NS4B in viral persistence?

Our hypothesis is included in the Discussion. Lines 1405-1486

Q: Section 3.4: It is quite puzzling that the antivirals work very well for the wild-type USUV, but do not work as well against the portion of the wild-type virus in the background of truncated genomes until higher doses are used.  The expectation would be that there are fewer wild-type genomes and the truncated forms cannot replicate efficiently without the help of the wild-type virus. What could be going on here?

We have repeated these experiments after suggestion by other referee to obtain more accurate data (IC50). We have found now that persistent virus is slightly more sensitive to mutagenic drugs than wild type virus during a lytic infection. As mentioned above, the overall message of our paper doesn’t change but it affects to the interpretation of the data. Consequently, we have modified substantially the manuscript on these paragraphs where we discuss these different susceptibilities. Lines 823-866, 872-885

Line 381: cells for...

Changed

Line 372: We initially...

Amended

Section 3.6: It is also puzzling that withdrawal of treatment results in rescue of USUV when it appeared that there were no more infectious particles. Could the authors stretch the treatment time beyond what is presented or they expect no improvement when the drugs are withdrawn?

From the intracellular RNA graph we can infer that viral RNA will become undetectable (<log 10 = 3) after >30 days of treatment. We consider this is a very interesting possibility but that needs long-term planning. In a previous study working with unrelated persistent norovirus infection in vivo we managed to drive virus to undetectable levels after 2-month treatment with favipiravir. We can't extrapolate what we observed in mice with norovirus to data shown here in a persistently-infected cell model for USUV. Nonetheless, we are interested in investigating if favipiravir and/or ribavirin can effectively cure USUV during prolonged treatment and how long does it take until complete loss of intracellular infectivity. This information can be of relevance to design improved therapies.

Line 423: ...monolayers for...

This has been changed

Q: Figure 4 and 5: Proliferation/toxicity assays should accompany these figures to indicate the level of toxicity associated with the compounds.

We have now included toxicity assays in Figure 4 (A, B, C, D) for Vero cells and also for persistently-infected cells. We found that persistently infected cells are modestly less affected by antiviral drugs.

Line 373: We initially...

Amended

Line 381: for 10 days...

Amended

Line 413: cells were treated...

Agreed

Line 423: monolayers for...

Agreed

Line 466: Are some bird species more susceptible than others and is there a possibility of developing an animal model of USUV persistence. The failure to completely destroy viral RNA following removal of drugs may require involvement of the adapative immune system.

Yes, there is different susceptibility to infection in different bird species. We have now included a sentence suggesting the possibility of developing animal models of USUV persistence. Lines 1237-1242

Line 474: Despite the lack of knowledge regarding...

Line 499: Can persistently infected Vero cells superinfected?

Although there are not many examples of superinfection exclusion for Vero cells, it has been demonstrated for infections with several viruses. Resistance to superinfection has been documented for Vero cells persistently infected with Murray Valley encephalitis virus (Poidinger et al 1991 Journal of General Virology (1991), 72, 573-578). Superinfection exclusion has also been found in cells infected with rubella virus (Claus et al. Journal of General Virology (2007), 88, 2769–2773).

Line 505: mainly affecting European...

Amended

Line 567: but the virus relapses when drugs are withdrawn!!!

We have now edited this part to make it more accurate

Reviewer 3 Report

Major comments

The data, which are presented in the paper, are interesting and the hypotheses of the authors are reasonable. However they should be presented in a more consistent way.

-       Why do you show the results for V cells and S cells? What is the difference for the conclusions? Wouldn´t it be enough to present just one (especially because there are only S cells shown in same figures)?

-       There is no obviously reason and nothing is mentioned why there are different passages used in the different antiviral experiments

-       Reduce the data of the antiviral experiments to the main message

-       Mention other antiviral studies where they used a drug concentration of 2000 µM

-       Does any of the mutations leads to a loss of infectivity for insects? Add experiments with insect cells to the end of the mutation analysis

Minor comments

-       Comparison to TBEV should be shifted to the discussion

-       Figure 1D > show data in another way

-       Line 226 > reword title

-       Figure 2B > Why do you show six different PCR products? Explain more in detail or show less (would be nice to have a constant number in all experiments, see 3 C+E)

-       Figure 3 C+E > data should be presented in a consistent way, see Figure 2B, here are five examples presented

-       Line 311 > explain/reword “repeatedly found”

-       Figure 5 > Why do you performed the experiments only with S cells for 2000 µM? If you show the results for both cells in all figures, including Fig. 5A, you must show also it also or make an explanation

-       Figure 4 > Why do you use different passages for S and V cells (p9 / p13)? Should be the same! And why don’t you use passage 40?

-       Figure 6 > Again: Why do you use this passage (p34)? And why don’t you use passage 40?

-       Figure 6+7+8 > Too much data, reduce this graphs and just show the main basics, which will support your hypothesis. For example: Why do you show Graph 6B? Exclude and just mention in the text

-       Figure 6+7+8 > This data are confusing. Why now passage 39? Be consistent in the number of days where you treat and remove drugs. Again: In Fig. six both cell lines shown, in Fig 7 only S cells, explanation is lacking

-       3.7 > Is their a reason why you used this supernatant (Fig. 7A) for infection? Explain

-       Figure 9 > Mention which cells are used for this experiment

-       Line 467 > explain in more detail how this supports the possibility

-       Line 471-473 > reword

-       Line 481-843 > show same examples

-       Line 474 ff > Is this your own hypothesis, is it new? Make it more clear!

-       Line 487-488 > How?

-       Line 498 > Be careful with the term replicon and DVG

-       523 > Norovirus doesn’t fit well as example, use a more USUV-related virus

Author Response

Dear reviewer #3,

We appreciate the time dedicated to review our manuscript and your comments made on the text. We deem your suggestions very valuable and consider that our revised manuscript is now much improved. We have now introduced extensive modifications to make our data clearer and to respond all your queries. In particular, we have carried out new antiviral analysis using the same cells in all the experiments which are now included in Figure 4 and Table 2. This information has permitted us to clarify that USUV in persistence is slightly more sensitive to nucleoside drugs than in lytic infections, in contrast with previous observations. This fact doesn’t affect the main message of our manuscript but we have now introduced some modifications to clarify these new observations.

Besides, in the revised version we have introduced the following changes:

-  Format modifications of some panels of Figures 1 and Figures 3

-  Figure 4 and 5 have been deleted and replaced with new data (Figure 4 and Table 2). These data include IC50, CC50 and selectivity index determinations

-  Figure 6 has been moved to Supplementary data (Figure S3)

-  New data has been included as supplementary information (Figures S1 and S2)

- Modification of the title to make it more accurate

Responses to specific comments are provided below

Reviewer 3

Q: Why do you show the results for V cells and S cells? What is the difference for the conclusions? Wouldn´t it be enough to present just one (especially because there are only S cells shown in same figures)?

The main goal of including two lines of persistent infection is to add robustness to our data. This has been especially relevant to interpret several results observed here. We found that V and S both show fluctuating titres along passages but without tendency to increases or decreases in titres (Figure 1C), both show a tendency to reduced specific infectivity (Figure 1E and 1F) and both viruses lead to similar truncations and the accumulation of 4 identical mutations. This reproducibility in two independent lineages supports that these are biologically relevant features to the establishment of persistence in cells. We have now included one sentence in lines 116-118 to clarify this.

Q: There is no obviously reason and nothing is mentioned why there are different passages used in the different antiviral experiments

We totally agree with the reviewer here, this is an awful mistake. We are embarrassed of not having anticipated this before. We have now removed Figures 4 and 5 and obtained new data using V and S cell lines from passage 39 (New Figure 4 and Table 2). This is in agreement with previous data in Figures 7, 8 and 9 using passage 39 (now these figures are numbers 5, 6, 7). Old figure 6 (work with passage 35) has now been transferred to Supplementary data. We consider this is a relatively close passage to 39 and the information is relevant to subsequent experiments.

Q: Reduce the data of the antiviral experiments to the main message

We agree with the reviewer in making this information more focused. We have reduced the number of figures on antiviral experiments from 6 to 4. We have also significantly modified the text.

Q: Mention other antiviral studies where they used a drug concentration of 2000 µM

We have now included data in vivo (non-human primates) and experimental procedures in vitro involving these concentrations. Lines 1088-1089, 1347-1351

Q: Does any of the mutations leads to a loss of infectivity for insects? Add experiments with insect cells to the end of the mutation analysis

We are extremely interested in examining how adaptation during persistence in mammalian cells affects to infectivity in insect cells. We are also very keen to assess the impact of antiviral mutagenesis in an invertebrate host context. Unfortunately, we work in a BSL unit at DTU which contains a single 37C incubator (shared with different groups). Hence, we haven’t yet had the opportunity to implement insect cell culture systems of infection here, and thus we are unable to perform these experiments in the short term.

We are currently in discussions to relocate our group to a different Institute where we expect having access to 30C incubators. This may enable us in these experiments proposed here. Unfortunately, if this finally happens it won't be before 2020. We agree with the reviewer that this is a fascinating field of study that we are very willing to pursue in the near future shall the facilities become available.

Minor comments

Q: Comparison to TBEV should be shifted to the discussion

We agree. We have moved comments on TBEV from results to discussion

Q: Figure 1D > show data in another way

We agree with the reviewer that our previous data was a bit messy. We are now representing specific infectivity in a much clearer way. Old Figure 1D is now represented in Figures 1E and 1F

Q: Line 226 > reword title

Amended

Q: Figure 2B > Why do you show six different PCR products? Explain more in detail or show less (would be nice to have a constant number in all experiments, see 3 C+E)

We apologise for having presented this data in a way that is misleading to a wrong interpretation. Figure 2 and 3 are representing different things and not directly connected to each other. In Figure 2 we have represented 6 different amplicons that cover the entire genome to attempt to delimit the genomic regions deleted in persistent viruses.

Two amplicons cover the entire genome: 1-7950 (I) and 6841-11066 (II). The remaining amplicons are shorter versions of the 5' end that help us identify the deleted region. Amplicons 1-4218 (III) and 1-3359 (IV) both contain deletions suggesting that the deleted region is between residues 1 and 3359. Amplicons, 818-3359 (V) and 1-2673 (VI) only show wild type virus. This means that residues 818-2673 are deleted in all defective genomes.

In Figure 3C and 3E however we are representing only qPCR data. There are only 2 PCR targets (Env and NS5). Different symbols in each figure represent independent biological replicas and NOT independent amplicons (e.g. cells seeded in 5-7 wells and RNA extracted in each replica individually). We agree that the data is not clear. We have now represented Figure 3 in a different way to avoid misunderstandings. We have also amended the text accordingly Lines 364-374, 607-735, 746-749.

Q: Figure 3 C+E > data should be presented in a consistent way, see Figure 2B, here are five examples presented

Please see explanation above. We have modified the text and Figure accordingly to make it clearer.

Q: Line 311 > explain/reword “repeatedly found”

We completely agree, we have modified this paragraph to avoid redundancies

Q:  Figure 5 > Why do you performed the experiments only with S cells for 2000 µM? If you show the results for both cells in all figures, including Fig. 5A, you must show also it also or make an explanation

We totally agree, the new data provided in Figure 4 include treatment of V cells with 2000 uM drugs

Q:  Figure 4 > Why do you use different passages for S and V cells (p9 / p13)? Should be the same! And why don’t you use passage 40?

We completely agree with the reviewer. This is a very embarrassing mistake. We have deleted old Figures 4 and 5. Figure 6 (work with passage 35) has now been moved to Supplementary data. We have repeated these experiments always using the same cells (passage 39) in data provided in the new Figure 4 and Table 2.

Q:   Figure 6 > Again: Why do you use this passage (p34)? And why don’t you use passage 40?

As this is a preliminary experiment and is a passage relatively close to p39 we have decided to maintain this data as a Supplementary Figure (Figure S3) but we agree that we must remove it from the main text.

Q: Figure 6+7+8 > Too much data, reduce this graphs and just show the main basics, which will support your hypothesis. For example: Why do you show Graph 6B? Exclude and just mention in the text

We agree this is much data. We have now moved Figure 6 to Supplementary data files. We have also modified extensively the text and make it more straightforward.

Q: Figure 6+7+8 > This data are confusing. Why now passage 39? Be consistent in the number of days where you treat and remove drugs.

We completely agree. As mentioned above, all our experiments with antivirals in the main text have been performed only with passage 39 cells.

Q: Again: In Fig. six both cell lines shown, in Fig 7 only S cells, explanation is lacking

We carried out a preliminary experiment where both cell lines were (apparently) cured in the presence of favipiravir and ribavirin (old Figure 6, now Figure S3). Owing to this similar sensitivity to treatment, we decided to focus on a single cell line to characterise the molecular dynamics of extinction. This experiment entailed a complex experimental set up, and thus we decided to work only with one cell lineage. We have now added further clarification in the main text (lines 1081-1087).

Q: 3.7 > Is their a reason why you used this supernatant (Fig. 7A) for infection? Explain

In this experiment we treated persistently infected cells with different drugs and analysed the antiviral effect on persistent virus (7B). These cells are already infected as they are carrying the virus persistently (see Figure 1C). Thus, we don’t infect these cells in 7B, we just treat them with drugs and collect supernatants at different time points. We analysed virus titres in these samples.

In 7C we carry out infections in normal Vero cells using supernatants collected from persistent infections in 7B. We carry out this blind passage in 7C to confirm that USUV is extinct in those samples where we find no infectivity in 7B. We confirmed virus extinction in all the samples collected in 7B showing no infectivity with the exception of supernatants of persistent cells treated with RBV during 6 days (7C). We have now edited the legend of Figure 7 to make it clearer to the readers. Please take into consideration that old Figure 7 is now Figure 5.

Q: Figure 9 > Mention which cells are used for this experiment

This information is now provided in the figure legend (Line 1190). Please note that old figure 9 is now figure 7.

Q: Line 467 > explain in more detail how this supports the possibility

We have now expanded on this hypothetical possibility in lines 1234-1237

Q: Line 471-473 > reword

We have now edited these sentences

Q: Line 481-843 > show same examples

We have now included several examples of truncated flavivirus genomes

Q: Line 474 ff > Is this your own hypothesis, is it new? Make it more clear!

This is in line with previous lines of evidence. We have now amended this sentence as suggested

Q: Line 487-488 > How?

We argue some few lines below (1333-1340) that DVGs may be protecting the cells from superinfection by a mechanism known as superinfection exclusion.

Q: Line 498 > Be careful with the term replicon and DVG

We have now indicated that this replicon is a synthetic transcript

Q: 523 > Norovirus doesn’t fit well as example, use a more USUV-related virus

We agree. We have included recent data with USUV in vivo

Round 2

Reviewer 1 Report

Revised version of manuscript viruses-479319 by Navarro Sampere and co-workers has been extensively modified and improved according to reviewer´s comments and suggestions, and new experiments have been performed to support initial conclusions. Overall, data are now better and more clearly present with additional information. As indicated by the authors in their responses, several times they apologize for gross mistakes in the initial submission. I recommend the authors to carefully checking their future articles before sending them for publication.

Author Response

Dear reviewer #1

We appreciate all the comments made on our manuscript. These have been very valuable to deliver now a much better work. We also deem that this experience constitutes good training to avoid making some mistakes pointing out by all the reviewers that we made during the preparation of the first version of this paper

Reviewer 3 Report

The authors changed major points of criticism and rearranged some of the data to make it clearer. 

Even if you are not able to perform experiments in insect cells you should discuss this point.

Minor comments

Line 106-108: reword sentence, ambiguous worded

Line 114-114: reword, the infection is apparently not totally lytic

Line 189: provide detailed information’s regarding the concentrations

Line 289: Figure 1B doesn´t show any percentages, see comments to Fig.1

Figure 1B: is the legend of the y-axis correct? Only % seems to be more reasonable

Line 295-300: this information is not mentioned in the discussion. Either you delete this paragraph or you add this information into your discussion

Line 326: change “in the” to “for other”

Figure 4E-H: Legend of “lytic” correct?

Line 740-742: Mention the concentration of 800 µM which was used in (45) and explain the difference to this study

Line 1039-1053: Reword the whole paragraph! Sentences are partly not finished and the sense is lost

Line 1054-1061: Make a context to the presented data

Line 1150: If you say “third hypothesis” you should also mention “first” and “second”

Figure S1B: Why do you show V P39 (day3)?  Inconsistent, you don´t show day 3 of S cells. Delete. There is not more information if you show 10x and 20x, show only one magnification.

Author Response

Dear reviewer #3

We want to take the opportunity to thank you again for all the constructive feedback provided. This information is very helpful to improve our manuscript. Please find below point-to-point responses to all your queries

Reviewer 3

The authors changed major points of criticism and rearranged some of the data to make it clearer.

Even if you are not able to perform experiments in insect cells you should discuss this point.

We have now included a short paragraph discussing this point at the end of the Discussion section

Minor comments

Line 106-108: reword sentence, ambiguous worded

Amended

Line 114-114: reword, the infection is apparently not totally lytic

Modified

Line 189: provide detailed information’s regarding the concentrations

Amended

Line 289: Figure 1B doesn´t show any percentages, see comments to Fig.1

Amended

Figure 1B: is the legend of the y-axis correct? Only % seems to be more reasonable

Amended

Line 295-300: this information is not mentioned in the discussion. Either you delete this paragraph or you add this information into your discussion

This has been shortened in the results section, and the old paragraph has been now included in the first paragraph of Discussion

Line 326: change “in the” to “for other”

Modified as requested

Figure 4E-H: Legend of “lytic” correct?

In this figure, we are referring to an infection with wild type USUV (lytic infection) for comparison purposes. We have now modified the legend to make it clearer

Line 740-742: Mention the concentration of 800 µM which was used in (45) and explain the difference to this study

Amended

Line 1039-1053: Reword the whole paragraph! Sentences are partly not finished and the sense is lost

Paragraph extensively modified (now second paragraph of discussion)

Line 1054-1061: Make a context to the presented data

We have provided a context to this paragraph (now 3rd paragraph discussion)

Line 1150: If you say “third hypothesis” you should also mention “first” and “second”

Amended

Figure S1B: Why do you show V P39 (day3)?  Inconsistent, you don´t show day 3 of S cells. Delete. There is not more information if you show 10x and 20x, show only one magnification.

Amended